# REV-ERBα integrates colon clock with experimental colitis through regulation of NF-κB/NLRP3 axis

Shuai Wang[1,2], Yanke Lin[1], Xue Yuan[1], Feng Li[3], Lianxia Guo[1] & Baojian Wu [1,2]

The roles of Rev-erbα and circadian clock in colonic inflammation remain unclarified. Here we show colon clock genes (including *Rev-erbα*) are dysregulated in mice with DSS-induced colitis. In turn, disruption of the circadian clock exacerbates experimental colitis. Rev-erbα-deficient mice are more sensitive to DSS-induced colitis, supporting a critical role of Rev-erbα in disease development. Further, Rev-erbα ablation causes activation of Nlrp3 inflammasome in mice. Cell-based experiments reveal Rev-erbα inactivates Nlrp3 inflammasome mainly at the priming stage. Rev-erbα directly represses Nlrp3 transcription through specific binding to the promoter region. Additionally, Rev-erbα represses p65 transcription and indirectly repressed Nlrp3 via the NF-κB pathway. Interestingly, Rev-erbα activation in wild-type mice by SR9009 attenuates DSS-induced colitis, whereas the protective effects are lost in *Nlrp3* $^{-/-}$ and *Rev-erbα* $^{-/-}$ mice. Taken together, Rev-erbα regulates experimental colitis through its repressive action on the NF-κB/Nlrp3 axis. Targeting Rev-erbα may represent a promising approach for prevention and management of colitis.

[1] Reserach Center for Biopharmaceutics and Pharmacokinetics, College of Pharmacy, Jinan University, Guangzhou 510632, China. [2] Guangdong Province Key Laboratory of Pharmacodynamic Constituents of TCM and New Drugs Research, Jinan University, Guangzhou 510632, China. [3] Guangzhou Jinan Biomedicine Research and Development Center, Jinan University, Guangzhou 510632, China. These authors contributed equally: Shuai Wang, Yanke Lin. Correspondence and requests for materials should be addressed to B.W. (email: bj.wu@hotmail.com)

Many aspects of physiology and behaviors in mammals are subjected to circadian rhythms (a 24-h oscillation)[1]. Disruption of circadian rhythms has been associated with various types of diseases such as cancers and metabolic disorders[2,3]. Circadian rhythms are driven by the mammalian clock systems that are organized in a hierarchical manner[4]. The central clock system (pacemaker), located in the suprachiasmatic nucleus of the hypothalamus, synchronizes peripheral clocks (present in peripheral organs) through neural and hormonal signals[4]. At the molecular level, circadian clock machinery consists of transcriptional activators [circadian locomotor output cycles kaput (CLOCK) and brain and muscle ARNT-like 1 (BMAL1), forming the positive limb] and repressors [e.g., PER (period) and CRY (cryptochrome), forming the negative limb][5]. CLOCK and BMAL1 function as a heterodimer that activates transcription of clock-controlled genes, including *PER* and *CRY*. Once reaching a threshold level, PER and CRY proteins inhibit the activity of CLOCK/BMAL1, thereby repressing their own expressions. This type of transcriptional–translational feedback loop system generates circadian oscillations of clock-controlled genes[6].

REV-ERBα/β (NR1D1/NR1D2) are two members of the nuclear receptor 1D subfamily, functioning as transcriptional repressors[7]. They repress transcription of target genes through specific binding to the response element (named "RevRE" or "REV-ERB response element") in gene promoter and subsequent recruitment of co-repressors such as nuclear receptor corepressor-1 and histone deacetylase 3[8]. REV-ERBα is a core component of circadian clockwork as its deletion causes disruptions to circadian rhythms in mice[9]. In fact, REV-ERBα repression of BMAL1 is an accessory feedback loop that consolidates the rhythms of circadian oscillators[8]. In addition to circadian genes, REV-ERBα regulates the expressions of metabolic genes, thereby integrating circadian rhythms with cell metabolism[10]. Therefore, it is not surprising that REV-ERBα has been implicated in control of various physiological processes, including cell differentiation, lipid metabolism, mitochondrial biogenesis, and inflammation, making it a potential therapeutic target for cancers, dyslipidemia, and inflammatory diseases[11–13].

Ulcerative colitis (UC), one of two major types of inflammatory bowel diseases (IBD) (the other is Crohn's disease), is an acute or chronic inflammation of the membrane that lines the colon[14]. UC is characterized by weight loss, diarrhea, rectal bleeding, and abdominal pain, affecting millions of people in the world[14,15]. Although the exact cause of UC is uncertain, activation of the mucosal immune system and consequent pathological cytokine production play a contributing role[16–18]. Dextran sulfate sodium (DSS) is frequently used to induce colitis in experimental animals to study the pathogenesis of UC because of model simplicity and high similarities with human UC[19,20]. Disruption of circadian rhythms is reported to increase the risks for developing IBD[21]. Circadian perturbance also has the potential to alter gut microbiota, potentially contributing to IBD pathogenesis[22–24]. However, the mechanisms for regulation of IBD and microbiota by circadian clock remain largely unknown.

NOD-like receptor family pyrin domain containing 3 (NLRP3) inflammasome is a large protein complex consisting of NLRP3, ASC, and caspase-1[25]. NLRP3 inflammasome plays a central role in innate immune responses to pathogen-associated molecular patterns (PAMPs) or danger-associated molecular patterns (DAMPs)[25–27]. Activation of NLRP3 inflammasome involves two sequential steps (i.e., priming and assembling) trigged by two signals[28]. The priming step trigged by the first signal (e.g., a PAMP such as lipopolysaccharides (LPS)) activates nuclear factor-κB (NF-κB) signaling and induces the transcription of pro-interleukin (IL)-1β and Nlrp3. The second signal (e.g., a DAMP such as adenosine triphosphate (ATP)) triggers several signaling pathways, including potassium efflux, generation of reactive oxygen species, and lysosomal damage that induce the assembly of Nlrp3 inflammasome using the three components[29–32]. Activation of NLRP3 inflammasome promotes the cleavage of caspase-1 and maturation and secretion of proinflammatory cytokines IL-1β and IL-18[28].

The mechanisms for regulation of colitis by circadian clock remain elusive. In this study, we investigate a potential role of the core clock component Rev-erbα in colitis regulation. We first established a close relationship between colon clock system and DSS-induced colitis, identifying Rev-erbα as a potential link between circadian rhythms and colitis. Further, we revealed a critical role of Rev-erbα in development of experimental colitis through regulation of NF-κB and Nlrp3 inflammasome activities. Rev-erbα directly repressed *Nlrp3* transcription via specific binding to a RevRE site in *Nlrp3* promoter. Additionally, Rev-erbα repressed transcription of *p65* (a subunit of NF-κB) and indirectly repressed Nlrp3 via the NF-κB pathway. Our data suggest Rev-erbα as a drug target for prevention and management of colitis.

## Results

**Dysregulated clock genes in mice with experimental colitis.** DSS-induced colitis caused persistent and genome-wide gene deregulation in mouse colon (Fig. 1a). As expected, the mRNA levels of inflammatory cytokines were upregulated in the colon (Supplementary Figure 1). Pathway analyses of differentially expressed genes (DEGs) revealed a significant enrichment in the circadian clock system in addition to the inflammation pathways, suggesting colitis-associated clock dysregulation (Fig. 1b and Supplementary Table 4). In fact, oscillations of core clock genes were blunted in mice with colitis (Fig. 1c and Supplementary Table 5). Dysregulation of clock genes (*Nr1d1/Rev-erbα*, *Clock*, *Bmal1*, *Per2*, *Cry1*, *Npas2*, *Nr1d2/Rev-erbβ*, *Rorα*, and *Dbp*) were further confirmed by quantitative polymerase chain reaction (PCR) analyses (Fig. 1d).

**Circadian clock disruption exacerbates experimental colitis.** We first examined the effects of jet lag (i.e., physiologic disruption of circadian clock) on the development of colitis. Jet-lagged mice were established with a jet lag schedule of 8 h light advance every 2–3 days following a published protocol[3,33], and confirmed by a wheel-running test (Supplementary Figure 2A). Mice were subjected to jet lag for 8 weeks before DSS feeding. Compared with normal mice, jet-lagged mice were much more sensitive to DSS-induced colitis as evidenced by the inflammation index values (i.e., weight loss, disease activity index (DAI), histopathological score, colon length, and myeloperoxidase (MPO)) (Supplementary Figure 2B–G). We also examined the effects of Bmal1 knockout (i.e., genetic disruption of circadian clock) on colitis development. Bmal1 knockout mice were generated using the CRISPR/Cas9 technique, and validated by wheel-running test, PCR genotyping, and expression profiling (Supplementary Figure 3A–D). Similar to jet lag, Bmal1 ablation sensitized mice to DSS-induced colitis (Supplementary Figure 3E-J). Compared to wild-type mice, Bmal1-deficient mice showed aggravated weight loss, increased DAI and MPO values, a higher histopathological score, and shorter colons (Supplementary Figure 3E-J).

**Rev-erbα ablation sensitizes mice to experimental colitis.** Disruption of circadian clock (under both situations of jet lag and Bmal1 knockout) led to marked downregulation of *Rev-erbα* in the colon (Fig. 2a, b). We also observed diminished expression of *Rev-erbα* in mice with experimental colitis (Fig. 1). Thus, we

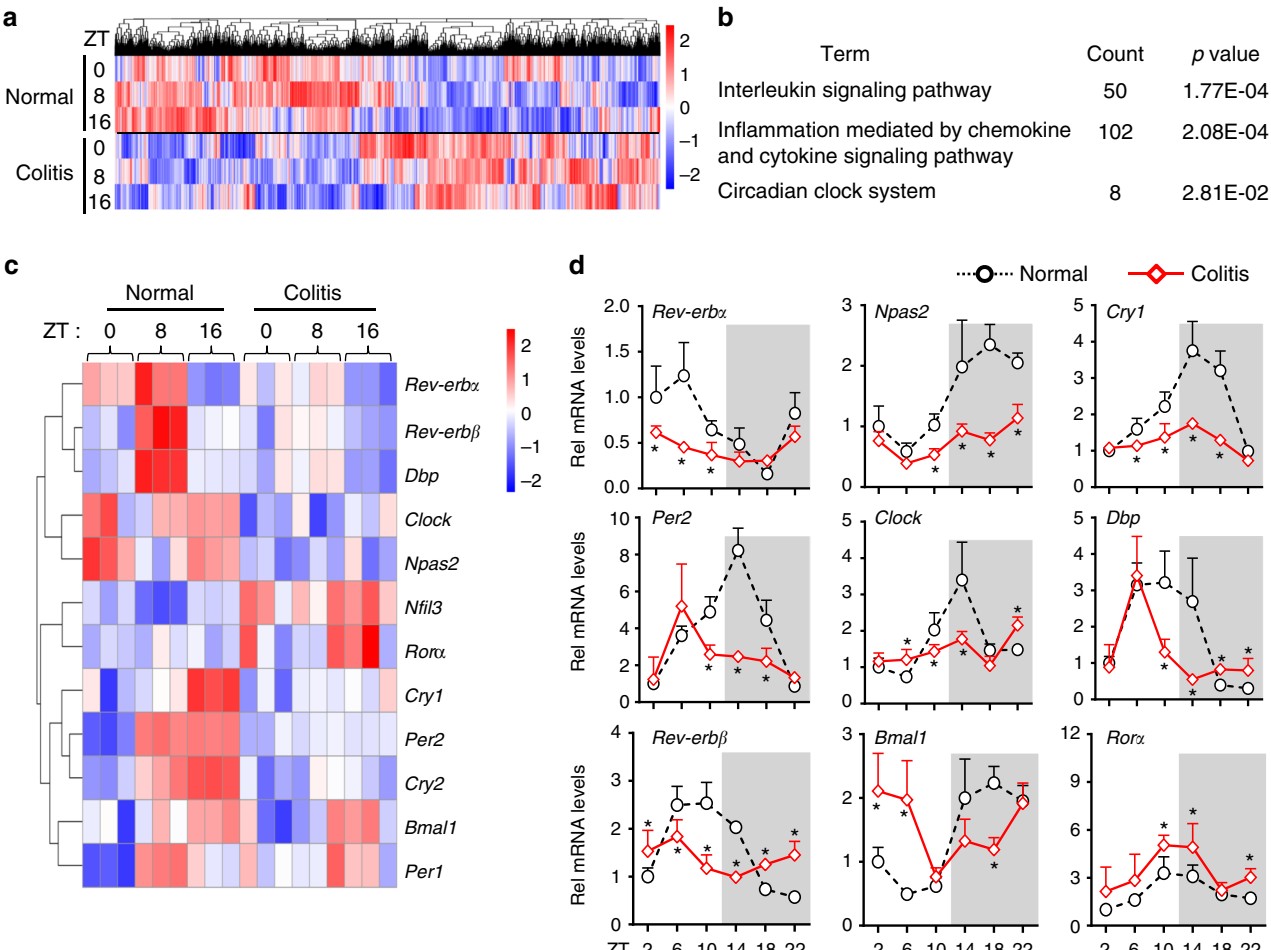

**Fig. 1** Circadian clock is dysregulated in mice with DSS-induced colitis. **a** Hierarchical clustering heatmap, comparing gene expressions in the colon between DSS-treated and control mice. Each column represents a gene and each row represents colon samples from different time points. Red indicates high relative expression and blue indicates low expression of genes as shown in the scale bar. **b** PANTHER pathway analysis results of differentially expressed genes (DEGs). **c** Heatmap of relative expressions of core clock genes in the colon. **d** qPCR assays of the core clock genes in DSS-treated and control mice. Data are presented as mean ± SD (n = 5). *P < 0.05 versus normal mice at individual time points (t test)

predicted a potential role of Rev-erbα in regulation of experimental colitis. This prediction was first interrogated by genetic studies. Rev-erbα knockout mice were generated using the CRISPR/Cas9 technique and validated by expression profiling (Fig. 3b and Supplementary Figure 4a–c). DSS-induced colitis was much more severe in Rev-erbα-deficient than in wild-type mice, supporting a critical role of Rev-erbα in the disease development (Fig. 2c–h). Further, IL-1β was the primary cytokine elevated in Rev-erbα$^{-/-}$ mice at the early phase of DSS-colitis (Fig. 2i). In addition, Rev-erbα ablation led to increased levels of colonic Nlrp3 proteins and IL-1β/IL-18 proteins (the products of Nlrp3 inflammasome activation) (Fig. 2j, k). Impact of Rev-erbα on Nlrp3 inflammasome activation was also confirmed using primary peritoneal macrophages (PMs) isolated from the genetic and wild-type mice (Fig. 2l). Inflammatory stimuli caused higher protein levels of Nlrp3, uncleaved and matured (cleaved) IL-1β in Rev-erbα$^{-/-}$ than in wild-type mice (Fig. 2l). Taken together, these data indicated an important role of Rev-erbα in regulation of Nlrp3 inflammasome and colitis development.

**Identification of *Nlrp3* as a clock-controlled gene.** Circadian expressions of *Nlrp3* and colitis-related inflammatory cytokines were determined in the liver and colon. In addition to the core

clock genes (e.g., *Bmal1* and *Rev-erbα*), hepatic *Nlrp3* displayed robust diurnal fluctuations (Fig. 3a and Supplementary Figure 5). Interestingly, *Nlrp3* oscillated in antiphase to *Rev-erbα* (a pattern highly similar to that of *Bmal1*, a direct target of Rev-erbα), suggesting that *Nlrp3* may be a target of Rev-erbα (Fig. 3a). *IL-1β*, *IL-18*, and *IL-6* (a known Rev-erbα target) showed mild oscillations (Fig. 3a and Supplementary Figure 5). Circadian expression of *Nlrp3* was also confirmed in the colons (Fig. 3b). However, the rhythmicity in *Nlrp3* expression was dampened as a result of Rev-erbα knockout (Fig. 3b). These data suggest that *Nlrp3* is a clock-controlled gene and a potential direct target gene of Rev-erbα.

**Rev-erbα inactivates Nlrp3 inflammasome.** Two treatment strategies alternating the order of SR9009 (a Rev-erbα agonist) and LPS were employed to explore the effects of Rev-erbα on Nlrp3 inflammasome. SR9009 prior to LPS treatment resulted in reduced Nlrp3 and IL-1β mRNAs and proteins in PMs, whereas SR9009 post LPS treatment showed no effects (Fig. 3c, d). Stimulation of PMs with LPS/ATP for a short period of time (30 min) led to activation of caspase-1 in PMs and SR9009 treatment had no effects on caspase-1 activation (Fig. 3e). These data indicated a main action of Rev-erbα on the priming rather than assembling step of Nlrp3 inflammasome.

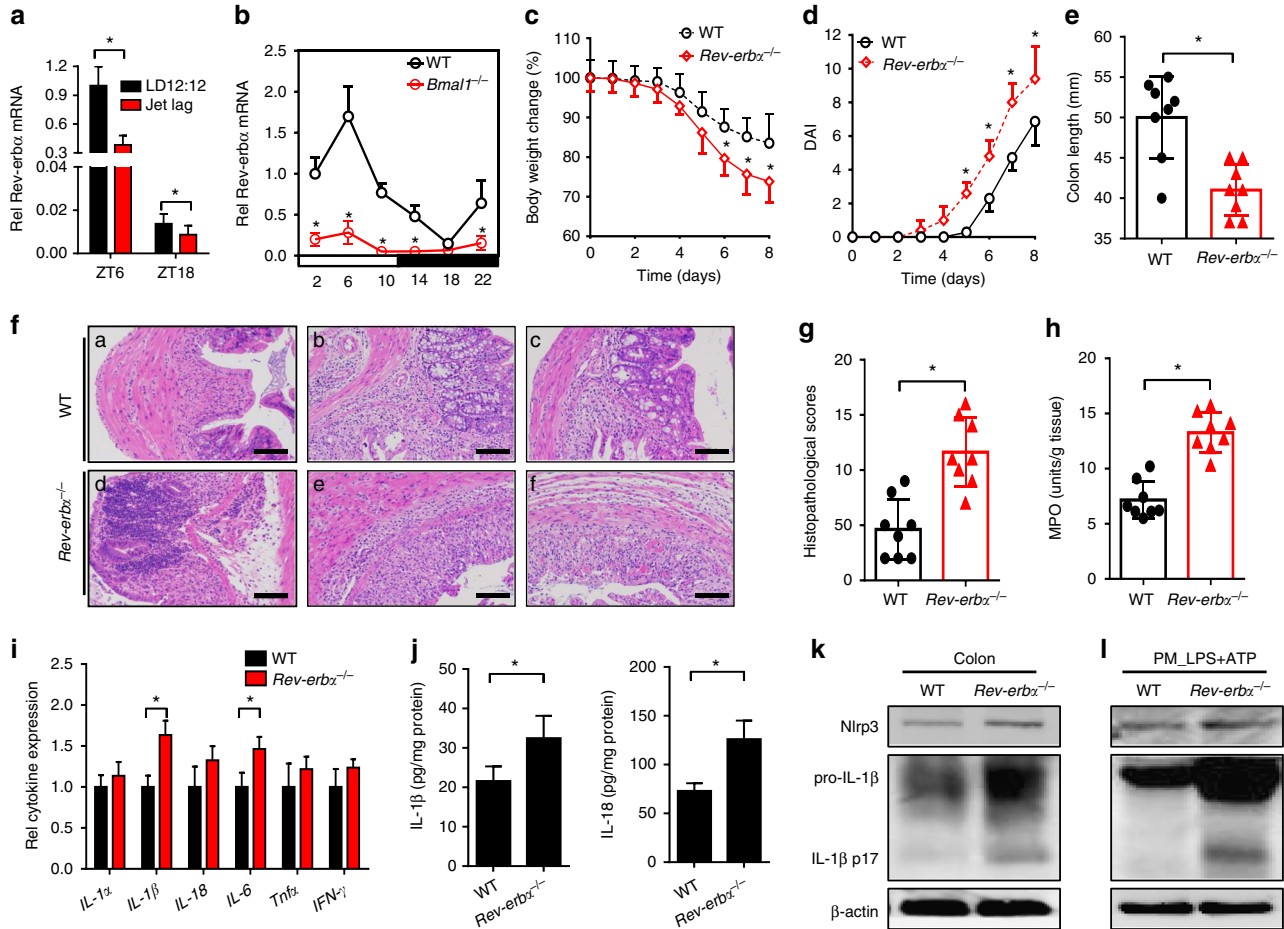

**Fig. 2** Rev-erbα ablation sensitizes mice to experimental colitis. **a** Circadian clock disruption by jet lag decreased Rev-erbα. Data are mean ± SD (n = 8). *P < 0.05 (t test). **b** Circadian clock disruption by Bmal1 deletion decreased Rev-erbα. **c** Weight loss measurements of wild-type (WT) and Rev-erbα⁻/⁻ mice treated with DSS. **d** DAI scores of wild-type and Rev-erbα⁻/⁻ mice treated with DSS. In panels **b**–**d**, data are mean ± SD (n = 8). *P < 0.05 versus WT at individual time points (t test). **e** Colon lengths of wild-type and Rev-erbα⁻/⁻ mice treated with DSS. Colon length was assessed at the time of necropsy. **f** Representative micrographs of H&E staining in the colon. Scale bar = 100 μm. **g** Histopathological scores of wild-type and Rev-erbα⁻/⁻ mice treated with DSS. **h** MPO activities of wild-type and Rev-erbα⁻/⁻ mice treated with DSS on day 8. **i** Expressions of inflammatory cytokines in colons of WT and Rev-erbα⁻/⁻ mice (2 days post DSS feeding) quantified by ELISA. Data are mean ± SD (n = 6). *P < 0.05 (t test). **j** ELISA measurements of colonic IL-1β or IL-18 on day 8 after DSS feeding. **k** Western blotting of Nlrp3, IL-1β, and β-actin in colons from WT and Rev-erbα⁻/⁻ mice on day 8 after DSS feeding. **l** Western blotting of Nlrp3, IL-1β, and β-actin in PMs from WT and Rev-erbα⁻/⁻ mice. PMs were stimulated with LPS for 8 h and ATP for the last 0.5 h. Each western blot is representative of three independent experiments (statistical differences between blot density levels were analyzed by Mann–Whitney U test, Supplementary Figure 12). For biochemical analyses, mice were sacrificed at ZT8 and colons were collected. In panels **e**, **g**, **h** and **j**, data are mean ± SD (n = 8). *P < 0.05 (t test). DAI: disease activity index, MPO: myeloperoxidase. Chemical concentrations: LPS (100 ng/ml), ATP (2 mM)

Rev-erbα agonists (i.e., SR9009, hemin, and GSK4112) dose-dependently attenuated LPS-induced Nlrp3 expression in Raw264.7 cells, indicating a repressive action of Rev-erbα on Nlrp3 and IL-1β expressions (Fig. 4a–d). By contrast, expressions of ASC and caspase-1 (the other two components of inflammasome assembly) were unaffected by Rev-erbα activation (Fig. 4a). Consistently, overexpression of Rev-erbα decreased Nlrp3 and IL-1β/IL-18 expressions, whereas knockdown of Rev-erbα enhanced their expressions (Fig. 4e, f and Supplementary Figure 6A). Moreover, SR9009 decreased caspase-1 p20 and mature (cleaved) IL-1β/IL-18 secretion from PMs co-treated with LPS/ATP, suggesting inactivation of Nlrp3 inflammasome by Rev-erbα (Fig. 4g and Supplementary Figure 6B). Likewise, SR9009 decreased Nlrp3 and IL-1β (both uncleaved and mature) proteins in bone marrow-derived macrophages (BMDMs) (Supplementary Figure 6C). Taken together, Rev-erbα blocked the priming stage of Nlrp3 inflammasome activation through repression of Nlrp3 expression.

**Rev-erbα represses Nlrp3 transcription**. A series of experiments were performed to explore whether Rev-erbα transcriptionally regulates Nlrp3. Additionally, we investigated a potential role of Rev-erbα/NF-κB axis in transcriptional regulation of Nlrp3 because Rev-erbα is a known repressor of NF-κB[34] and NF-κB is a transcriptional activator of Nlrp3[35]. In luciferase reporter assays with Raw264.7 and HEK293 cells, Rev-erbα repressed the transcription activity of Nlrp3 (Fig. 5a, b). Promoter analysis of Nlrp3 revealed that a 210-bp region (−1310 to −1100 bp) was responsible for the repressor activity of Rev-erbα (Fig. 5c). In silico prediction suggested one RevRE (Rev-erbα response element) and two κB sites (NF-κB-binding sites) within the 210-bp region (Fig. 5d, e). The two κB sites have been validated in a previous study[35]. Mutation of either RevRE (−1139/−1129 bp) or κB sites attenuated but failed to abolish the transcriptional repression effect of SR9009 (Fig. 5d). A mutation of both RevRE and κB sites completely abrogated the effects of SR9009 (Fig. 5d).

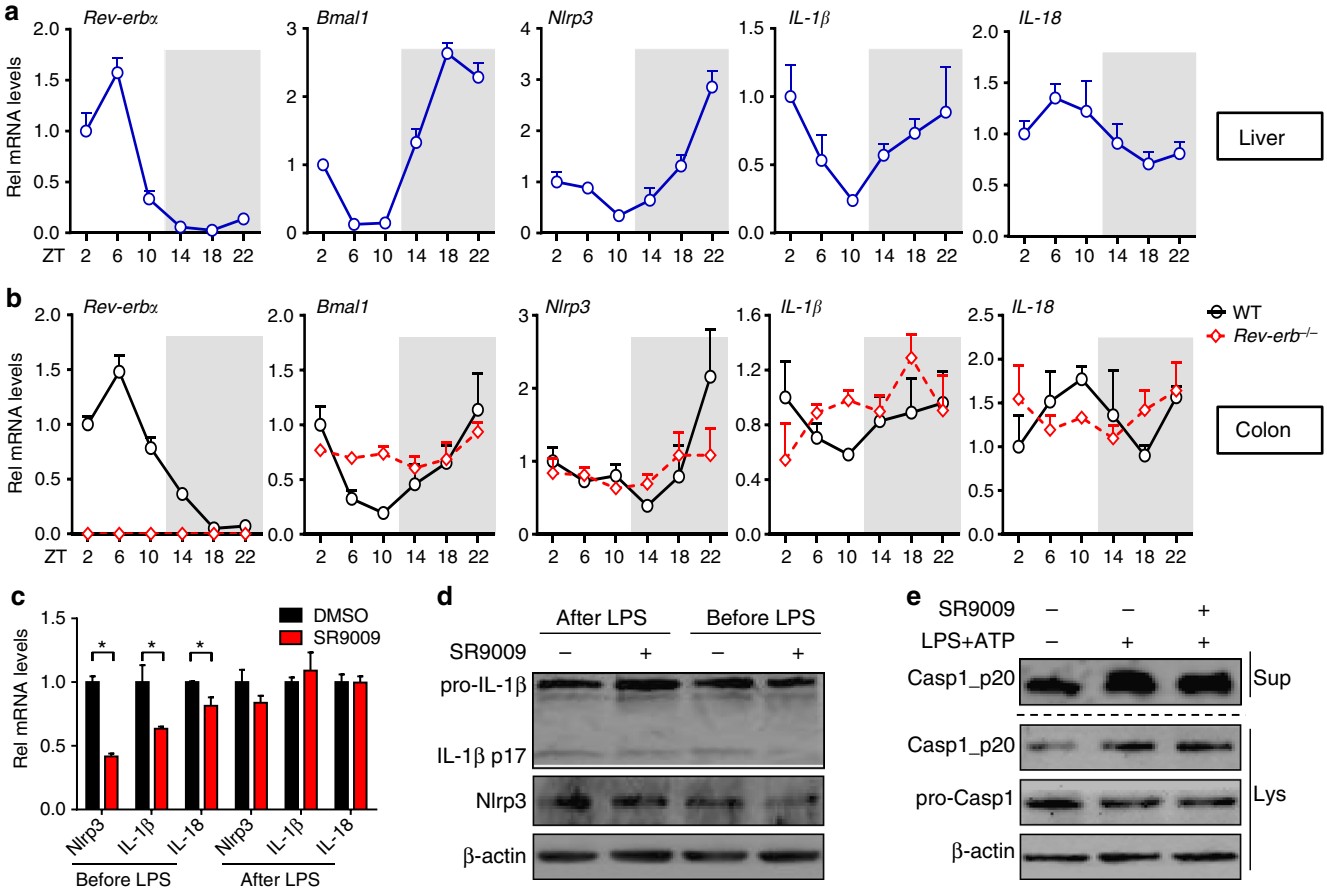

**Fig. 3** Identification of *Nlrp3* as a clock-controlled gene. **a** qPCR assays on circadian gene expressions of lives from WT mice. Data are mean ± SD (n = 5). **b** qPCR assays on circadian expressions of colons from WT and *Rev-erbα*[−/−] mice. Data are mean ± SD (n = 5). **c** qPCR measurements of *Nlrp3* and related genes in PMs after co-treatment of SR9009 (for 8 h) and LPS. LPS was added before or after SR9009 treatment. Data are mean ± SD (n = 3). *P < 0.05 (Mann–Whitney U test). **d** Western blotting of PMs after co-treatment of SR9009 (for 12 h) and LPS/ATP. LPS was added before or after SR9009 treatment for 3 h, followed by ATP addition for 30 min (added last). **e** Western blotting of PMs after treatment of SR9009 and LPS/ATP. PMs were pretreated with SR9009 or vehicle for 1 h, and then stimulated with LPS/ATP for 0.5 h. Each western blot is representative of three independent experiments (statistical differences between blot density levels were analyzed by Mann–Whitney U test, Supplementary Figure 12). The concentrations of SR9009, LPS, and ATP for cell treatment were 10 μM, 100 ng/ml, and 2 mM, respectively

The data suggested that both RevRE and κB sites were responsible for Rev-erbα-mediated repression of Nlrp3. Electrophoretic mobility shift assay (EMSA) assays indicated direct binding of Rev-erbα to the predicted RevRE site of *Nlrp3* (Supplementary Figure 7A). Further, chromatin immunoprecipitation (ChIP) assays confirmed recruitment of Rev-erbα to the RevRE site of *Nlrp3*, supporting direct interactions of Rev-erbα with *Nlrp3* promoter in vivo (Fig. 5f).

Next, we assessed the effects of Rev-erbα activation on NF-κB signaling. SR9009 treatment markedly suppressed LPS-induced IKBα phosphorylation as well as phosphorylation of p65 (a NF-κB subunit) in Raw264.7 cells, indicating an inhibitory action of Rev-erbα on NF-kB signaling (Fig. 6a). Inactivation of NF-kB signaling by Rev-erbα was supported by decreased mRNA expressions of *IL-1β*, *IL-18*, and *TNFα* (Fig. 4a and Supplementary Figure 6A & D), and also confirmed by reporter assay, EMSA, and immunofluorescence confocal microscopy (Fig. 6b, c, e, f). Further, total, cytosolic and nuclear p65 proteins were reduced by SR9009, potentially accounting for inactivation of NF-kB signaling (Fig. 6 a, d). Consistently, the mRNA level of *p65* in Raw264.7 cells was Rev-erbα-dependent and Rev-erbα over-expression decreased both p65 mRNA and protein (Fig. 6g and Supplementary Figure 8A). These data indicated a repressive role

of Rev-erbα in p65 expression and NF-κB signaling. By performing luciferase reporter, EMSA, and ChIP assays, we showed that Rev-erbα repressed p65 expression via its specific binding to a RevRE element (−474/−484 bp) in promoter region (Fig. 6h, i, Supplementary Figure 7B & 8B). Although Rev-erbα regulates *p65*, it shows no effects on *p50*, the other subunit of NF-κB (Supplementary Figure 8A).

**Rev-erbα activation alleviates experimental colitis**. SR9009 (or vehicle) was administered to mice once daily for 7 days prior to DSS challenge. Compared with vehicle-treated mice, SR9009-treated mice showed reduced body weight lost, a lower DAI score, a higher survival rate, and longer colons (Fig. 7a–d). Improvement of colonic inflammation was also evidenced by decreased histological score and MPO activity (Fig. 7e–g). These data suggested alleviation of DSS-induced colitis by Rev-erbα activation. Consistently, SR9009 failed to alleviate DSS-induced colitis in Rev-erbα-deficient mice (Supplementary Figure 9). Additionally, SR9009 administration post DSS challenge showed moderate protective effects on body weight loss (Supplementary Figure 10C). Further, SR9009 suppressed Nlrp3 inflammasome activation by

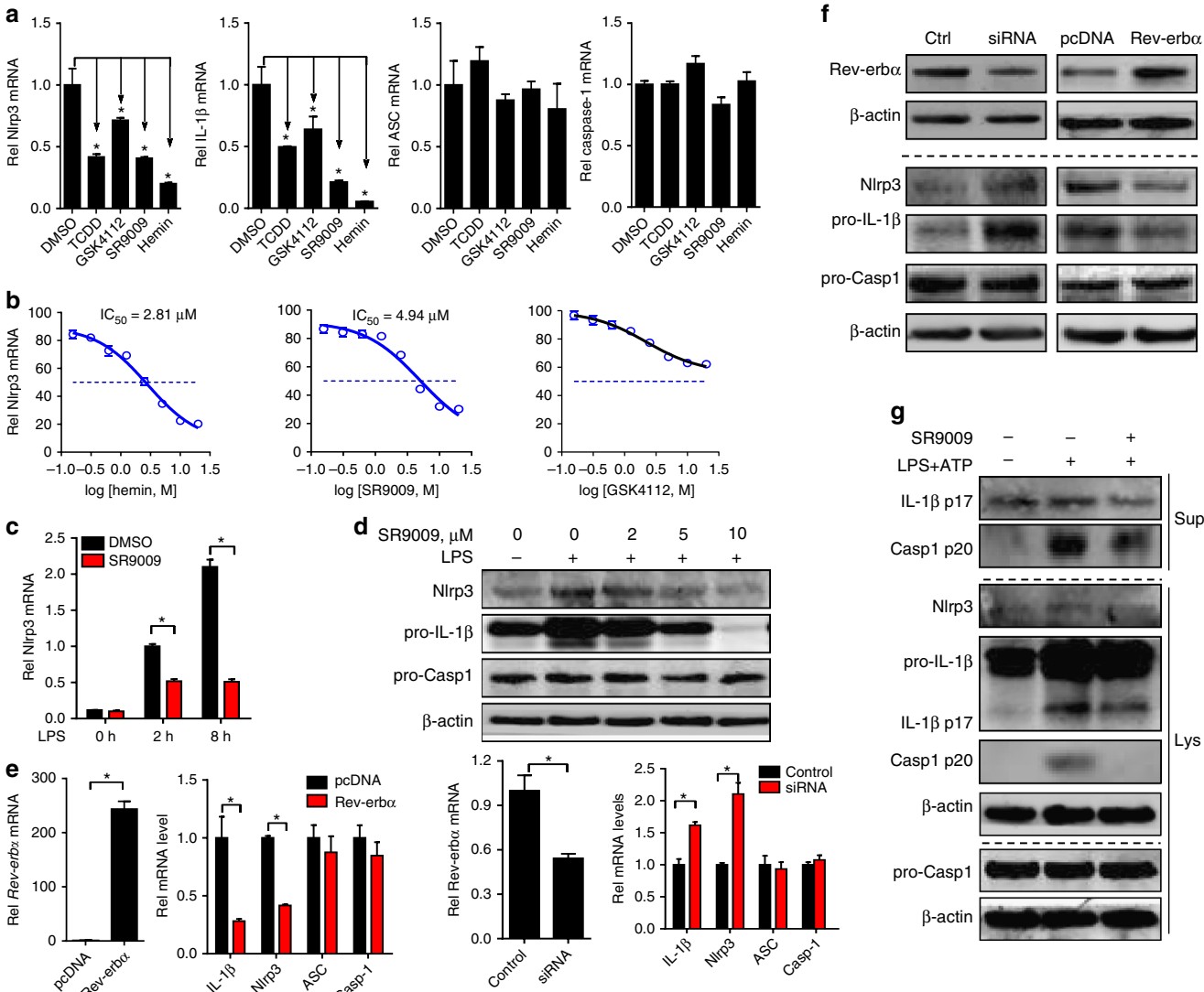

**Fig. 4** Rev-erbα inactivates Nlrp3 inflammasome at the priming step. **a** qPCR measurements of *Nlrp3, IL-1β, ASC*, and *caspase-1* in Raw264.7 cells. The cells were pretreated with Rev-erbα agonist (10 μM), TCDD (10 nM, as a positive control), or vehicle for 1 h and then stimulated with LPS for 8 h. Data are mean ± SD (*n* = 3). \*P < 0.05 versus DMSO-treated group (Mann–Whitney *U* test). **b** Measurements of *Nlrp3* mRNAs in Raw264.7 cells by qPCR. The cells were pretreated with DMSO or a series of concentrations of Rev-erbα agonist (SR9009, hemin, or GSK4112) for 1 h and then stimulated with LPS for 8 h. **c** Measurements of *Nlrp3* mRNAs in Raw264.7 cells by qPCR. The cells were pretreated with DMSO or SR9009 (10 μM) for 1 h and then stimulated with LPS for indicated hours. **d** Protein expressions of Nlrp3, pro-IL-1β, ASC, and pro-caspase-1 in Raw264.7 cells measured by western blotting. The cells were pretreated with DMSO or SR9009 (10 μM) for 1 h and then stimulated with LPS for 12 h. **e** mRNA expressions of *Nlrp3, IL-1β, ASC*, and *caspase-1* in Raw264.7 cells measured by qPCR. The cells were transfected with Rev-erbα siRNA or Rev-erbα plasmid for 24 h and then stimulated with LPS for 8 h. **f** Protein expressions of Nlrp3, Pro-IL-1β, ASC, and pro-caspase-1 in Raw264.7 cells measured by western blotting. The cells were transfected with Rev-erbα siRNA or Rev-erbα plasmid for 24 h and then stimulated with LPS for 12 h. **g** Western blotting of Nlrp3, cleaved caspase-1 (p20), cleaved IL-1β (p17), pro-caspase-1, and pro-IL-1β in the supernatants (Sup) or cell lysates (Lys) of PMs. PMs were pretreated with SR9009 or vehicle for 1 h and then stimulated with LPS for 12 h and ATP for 30 min (added last). The concentrations of LPS and ATP for cell treatment were 100 ng/ml and 2 mM, respectively. Each western blot is representative of three independent experiments (statistical differences between blot density levels were analyzed by Mann–Whitney *U* test, Supplementary Figure 12). In panels **c** and **e**, data are mean ± SD (*n* = 3), \*P < 0.05 (Mann–Whitney *U* test). Casp1: caspase-1

decreasing expressions of Nlrp3 and pro-IL-1β/pro-IL-18, reducing the formation of mature (cleaved) IL-1β/IL-18 in both PMs and colon (Fig. 7h–j). Contrasting with wild-type mice, Nlrp3-deficient mice were resistant to DSS-induced colitis (Fig. 7a–g). The protective effect of SR9009 on DSS challenge was lost in *Nlrp3*[−/−] mice (Fig. 7a–g), so were the repressive effects of SR9009 on IL-1β/IL-18 expressions (Fig. 7h–j). Taken together, Rev-erbα activation prevented DSS-induced colitis via suppression of NF-κB and inactivation of Nlrp3 inflammasome.

## Discussion

In this study, we first established a tight association between colon clock and experimental colitis. Based on loss-of-function studies, we showed the core clock component Rev-erbα was crucial in development of experimental colitis. Further, in vivo and in vitro experiments demonstrated that Rev-erbα inactivated Nlrp3 inflammasome by repressing NF-κB and Nlrp3 transcription, thereby integrating the clock clockwork with the colonic inflammation. It is noteworthy that Rev-erbβ, the paralog of Rev-erbα, did not show regulatory effects on Nlrp3 or p65

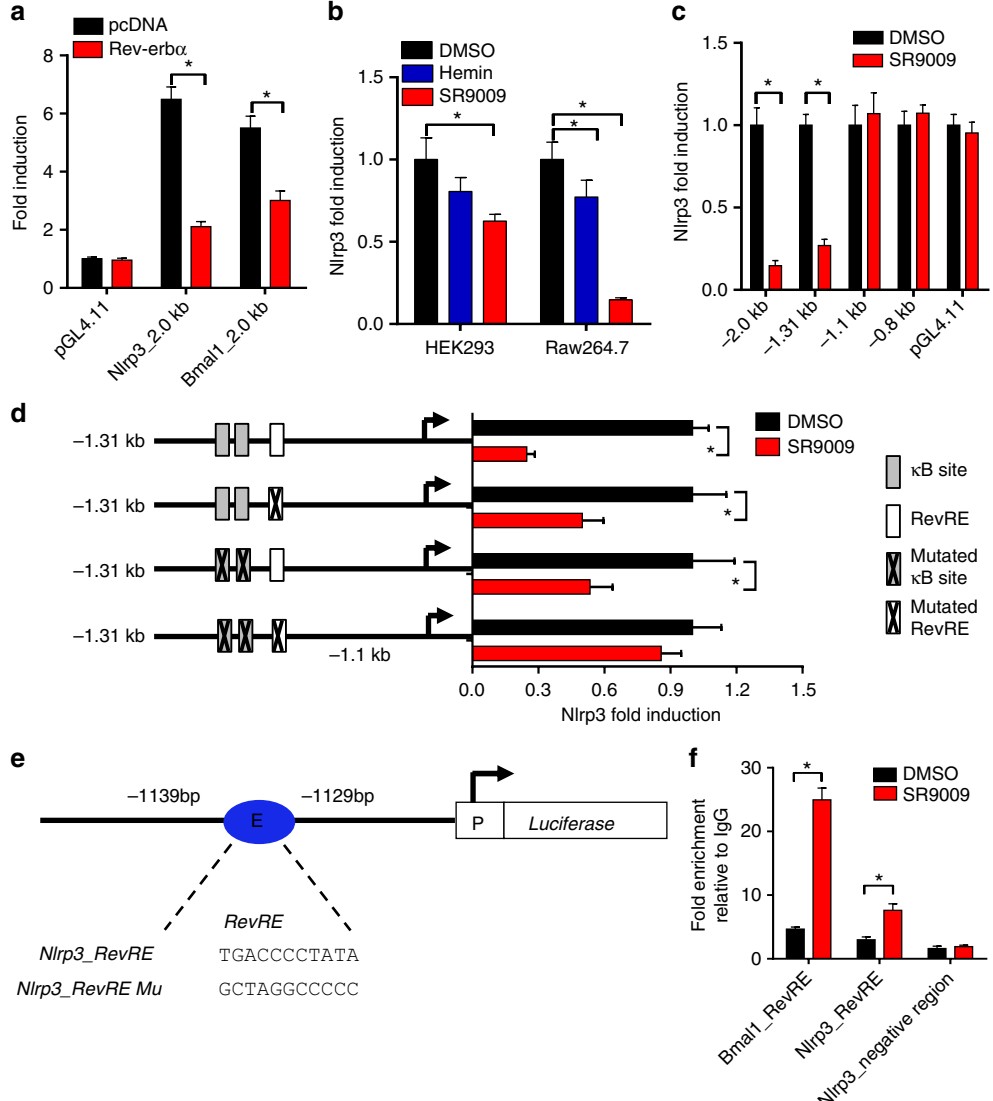

**Fig. 5** Rev-erbα binds to *Nlrp3* promoter and inhibits *Nlrp3* transcription. **a** Luciferase reporter assays showing that Rev-erbα represses *Nlrp3* transcription in Raw264.7 cells. The cells were transfected with blank PGL4.11, 2.0 kb *Nlrp3* reporter, or 2.0 kb *Bmal1* reporter along with blank pcDNA or Nr1d1 plasmid. Data are mean ± SD ($n = 4$). **b** Luciferase reporter assays showing that Rev-erbα activation downregulates *Nlrp3* transcription in HEK293 and Raw264.7 cells. HEK293 cells were transfected with the 2.0 kb *Nlrp3* reporter followed by treatment with SR9009 (or vehicle) for 12 h. Raw264.7 cells were transfected with the 2.0 kb *Nlrp3* reporter followed by treatment with SR9009 (or vehicle) for 1 h and LPS for 8 h. Data are mean ± SD ($n = 6$). **c** Luciferase reporter assays with distinct *Nlrp3* reporters in Raw264.7 cells. Raw264.7 cells were transfected with blank PGL4.11 and *Nlrp3* reporter (i.e., −2.0 kb, −1.31 kb, −1.1 kb, or −0.8 bp promoter reporter) followed by treatment with SR9009 (or vehicle) for 1 h and LPS for 8 h. Data are mean ± SD ($n = 6$). **d** Luciferase reporter assays with different versions of 1.31 kb *Nlrp3* reporters in Raw264.7 cells. The cells were transfected with 1.31 kb *Nlrp3* reporter or its mutated version followed by treatment with SR9009 (or vehicle) for 1 h and LPS for 8 h. Data are mean ± SD ($n = 6$). **e** Schematic diagram of *Nlrp3* luciferase plasmids with normal or mutated *Nlrp3* promoter. **f** ChIP assay, showing recruitment of Rev-erbα to *Bmal1* promoter and *Nlrp3* promoter in PMs. PMs were pretreated with SR9009 or vehicle for 1 h, followed by treatment with LPS for 1 h. Data are mean ± SD ($n = 4$). *$P < 0.05$ (t test or Mann–Whitney $U$ test). The concentrations of SR9009, LPS, and ATP for cell treatment were 10 μM, 100 ng/ml, and 2 mM, respectively

(Supplementary Figure 11). Interestingly, pharmacological activation of Rev-erbα by a small molecule (SR9009) protected mice from experimental colitis via a suppressive action on Nlrp3 inflammasome activity. Therefore, our study identified a mechanism for prevention and management of colitis. Further works are needed to establish optimal dose and dosing time for SR9009 in terms of drug development and clinical therapeutics.

Our data suggest Rev-erbα as a potential gatekeeper of intestinal inflammation. Rev-erbα reduces the severity of colitis in mice by repressing NF-κB and Nlrp3 expression, thereby downregulating Nlrp3 inflammasome activity (Figs. 4–6). At the time of manuscript preparation, Pourcet et al.[36] reported that Rev-erbα

regulates circadian expression of Nlrp3 and Rev-erbα activation alleviates fulminant hepatitis in mice. This study and the present one consistently pinpoint a critical role of Rev-erbα/Nlrp3 axis in controlling inflammatory diseases[36]. Both studies show that Rev-erbα directly represses Nlrp3 expression via its specific binding to a RevRE site (at the precise position of −1139/−1129 bp identified herein) within the promoter (Fig. 5d)[36]. However, we additionally demonstrated that Rev-erbα indirectly represses Nlrp3 expression via the transcription factor NF-κB (Figs. 5 and 6). Both direct and indirect regulation mechanisms play important roles in Rev-erbα repression of Nlrp3 because a mutation of both RevRE and κB site abolishes the repression effect of SR9009 on

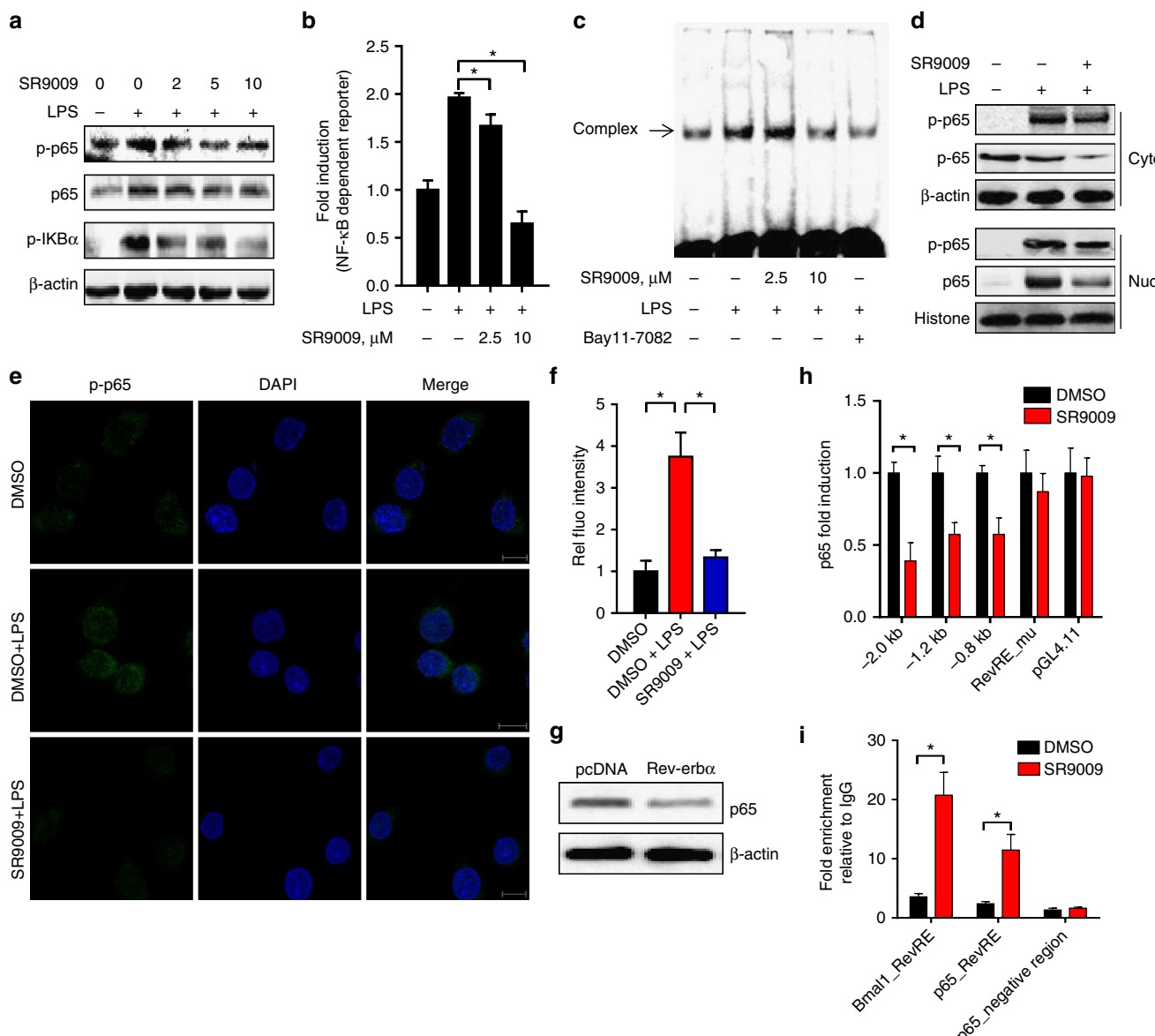

**Fig. 6** Rev-erbα activation inhibits NF-κB signaling and p65 transcription. **a** Protein expressions of p65, p-p65, and p-IKBα in Raw264.7 cells measured by western blotting. Cells were pretreated with indicated concentrations of SR9009 or vehicle for 1 h and then stimulated with LPS for 12 h. **b** Luciferase reporter assays with Raw264.7 cells, showing Rev-erbα-dependent activation of NF-κB pathway. Cells were transfected with the NF-κB-dependent reporter (containing four κB sites) followed by treatment with SR9009 or vehicle for 1 h and stimulated with LPS for 8 h. Data are mean ± SD ($n = 6$). *$P < 0.05$ versus the group treated with LPS alone ($t$ test). **c** EMSA assays performed with biotin-labeled NF-κB probe (containing NF-κB consensus binding sequence) in the presence of nuclear extracts. Raw264.7 cells were pretreated with Bay11-7082 (10 μM) or SR9009 for 1 h and then stimulated with LPS for 8 h. **d** Protein expressions of p65, p-p65, β-actin, and Histone H3 in cytoplasm (Cyto) or nucleus (Nuc) of Raw264.7 cells measured by western blotting. Cells were pretreated with SR9009 or vehicle for 1 h and then stimulated with LPS for 12 h. **e** Immunofluorescence analysis of p-p65 localization. Scale bar = 10 μm. **f** Intensity levels of green fluorescence for p-p65 quantified in six different fields. Data are mean ± SD ($n = 6$). **g** Effects of Rev-erbα overexpression on p65 protein in Raw264.7 cells. Each western blot is representative of three independent experiments (statistical differences between blot density levels were analyzed by Mann–Whitney $U$ test, Supplementary Figure 12). **h** Luciferase reporter assays in Raw264.7 cells with distinct *p65* promoter reporters. Cells were transfected with blank PGL4.11 and *p65* reporter [−2.0 kb, −1.2 kb, −0.8 bp, or RevRE (−474/−484 bp)_mutant], followed by treatment with SR9009 or vehicle for 1 h and stimulation with LPS for 8 h. Data are mean ± SD ($n = 6$). **i** ChIP assay with PMs, showing recruitment of Rev-erbα to *Bmal1* or *p65* promoter. PMs were treated with SR9009 or vehicle. Data are mean ± SD ($n = 4$). In panels **f**, **h** and **i**, *$P < 0.05$ ($t$ test or Mann–Whitney $U$ test). Concentrations of SR9009, LPS, and ATP for cell treatment were 10 μM, 100 ng/ml, and 2 mM, respectively

Nlrp3 expression, whereas mutations of either site cannot (Fig. 5d).

We observed increased IL-1β and IL-6 expressions in Rev-erbα-deficient mice at early stage of colitis (2 days of DSS feeding) (Fig. 2i). This agrees well with previous studies in which Rev-erbα represses transcription of IL-1β and IL-6[34,36]. Consistent with

Rev-erbα regulation, circadian expression of *IL-1β* and *IL-6* showed a typical pattern of Rev-erbα target gene (e.g., *Bmal1*) (Fig. 3a). We argue for a much more important role of IL-1β in development of colitis compared to IL-6. First, IL-1β was the primary inflammatory cytokine altered the most in Rev-erbα-deficient mice at the early phase of DSS-colitis (Fig. 2i). Second,

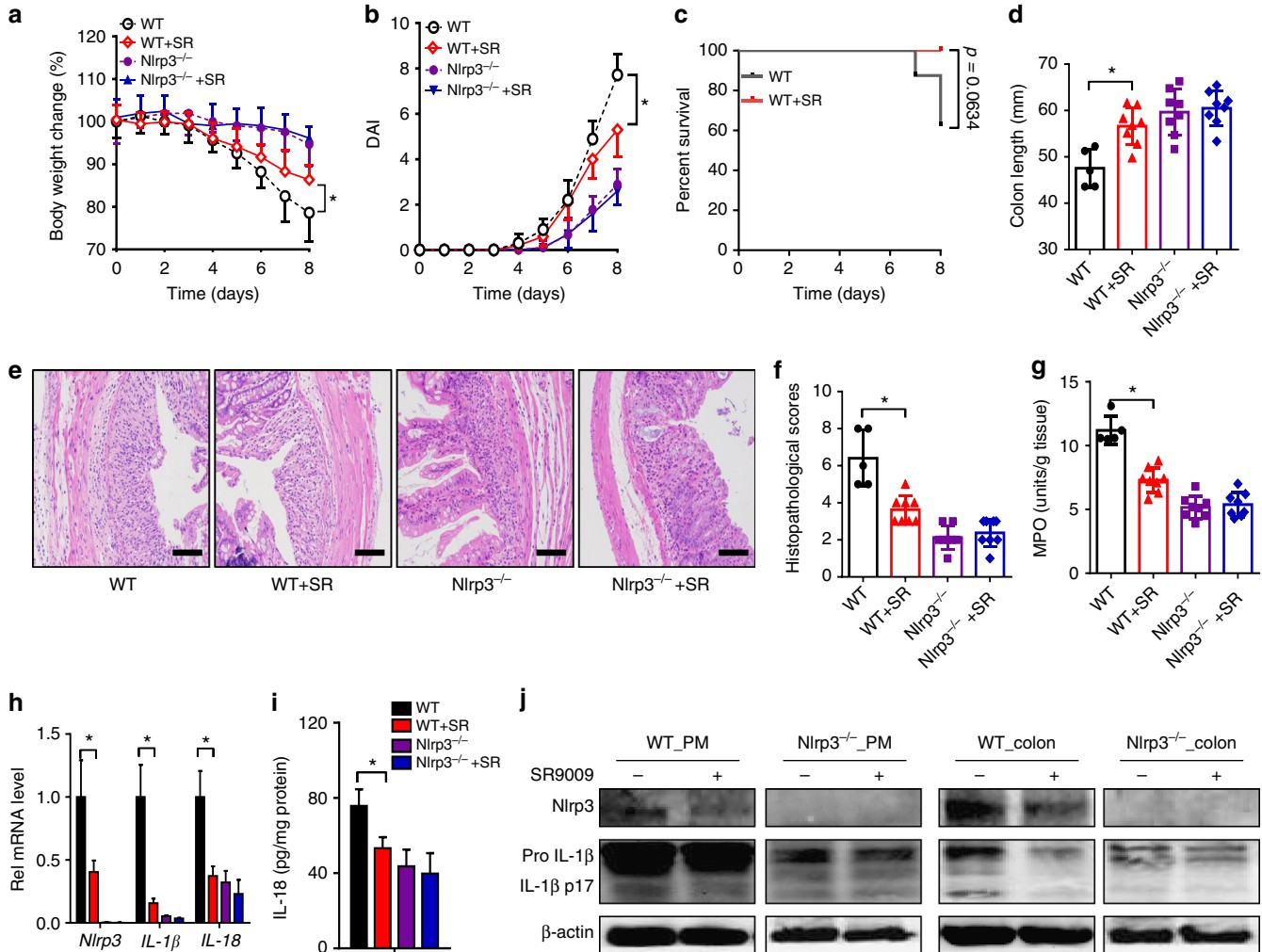

**Fig. 7** Rev-erbα activation alleviated experimental colitis in mice. **a** Weight loss measurements of four groups of mice with DSS feeding. **b** DAI scores of four groups of mice with DSS feeding. In panels **a** and **b**, data are mean ± SD ($n = 8$). *$P < 0.05$ ($t$ test). **c** Survival rates of SR9009-treated and control mice (log-rank test). **d** Colon lengths of four groups of mice treated with DSS. Colon length was assessed at the time of necropsy. **e** Representative micrographs of colon H&E staining. Scale bar = 100 μm. **f** Histopathological scores of four groups of mice treated with DSS. **g** MPO activities of mice colons on day 8. **h** qPCR analyses of Nlrp3, IL-1β, and IL-18 expressions in whole colon tissues of mice with colitis on day 8. **i** ELISA measurements of colonic IL-18 protein on day 8 after DSS feeding. **j** Western blotting of Nlrp3, IL-1β, and β-actin in PMs or colons from mice with colitis on day 8. Each western blot is representative of three independent experiments (statistical differences between blot density levels were analyzed by Mann–Whitney $U$ test, Supplementary Figure 12). SR9009 (50 mg/kg) was administered to mice via intraperitoneal injection once daily at ZT8 for 7 days prior to DSS treatment, and SR9009 dosing was continued along with DSS treatment. In panels **d**, **f–i**, data are mean ± SD ($n = 5$ for WT, $n = 8$ for other groups). *$P < 0.05$ ($t$ test or Mann–Whitney $U$ test). SR: SR9009

the regulatory effect of Rev-erbα on intestinal inflammation was lost upon deletion of Nlrp3 in mice (Fig. 7).

We provide strong evidence that Rev-erbα regulates activation of Nlrp3 inflammasome mainly at the priming stage. First, treatment of SR9009 + LPS decreased Nlrp3 and IL-1β expressions in PMs, whereas treatment of LPS + SR9009 showed no effects (Fig. 3c, d). Second, rapid stimulation of PMs with LPS + ATP led to caspase-1 activation (independent on Nlrp3 expression) that was unaffected in the presence of SR9009 (Fig. 3e). Third, Rev-erbα activation did not affect the expressions of ASC and caspase-1 (Fig. 4). Fourth, Rev-erbα was identified as a negative regulator of NF-κB activation and signaling that plays a key role in the priming of Nlrp3 inflammasome[37]. In the study of Pourcet et al.[36], the authors observed an increased number of cells with ASC specks in Nr1d1-deficient mice, thus proposed that Rev-erbα modulates the formation (assembling) of Nlrp3

inflammasome complex though the regulatory mechanism was unexplored. Whether Rev-erbα regulates assembly of Nlrp3 inflammasome complex awaits further investigations.

In line with a previous study[36], Rev-erbα represses the expressions of IL-1β and IL-18 in addition to Nlrp3 (Fig. 4e and Supplementary Figure 6A). The rhythms of colonic IL-1β and IL-18 mRNAs were also altered upon Rev-erbα deletion (Fig. 3b). This led to the speculation that downregulation of IL-1β and IL-18 mRNAs (as well as pro-IL-1β and pro-IL-18 proteins) also plays a role in Rev-erbα repression of colonic inflammation. However, we believe that contributions of pro-IL-1β and pro-IL-18 downregulation to colitis are none or negligible because impact of Rev-erbα on inflammasome activation (i.e., activation of caspase-1 and maturation of IL-1β) and experimental colitis is highly Nlrp3-dependent (Fig. 7).

Our results suggest a tight interconnection between circadian clock and immune system consistent with the literature[38,39]. Clock disturbance sensitized mice to experimental colitis by upregulating Nlrp3 inflammasome activity via the Rev-erbα receptor. Accordingly, Rev-erbα activation restores clock functionality and reduces the severity of colitis (Fig. 7). On the other hand, experimental colitis led to perturbed clock in the colon (Fig. 1). However, why this occurred was not addressed in current study. Nevertheless, control of circadian clock by immune system is highly possible because the immune mediators (e.g., cytokines) were shown to have strong effects on circadian rhythms[40,41]. In addition, DSS and SR9009 may modify the circadian behaviors[42,43]. Whether and how such circadian alterations affect inflammatory processes remain unexplored.

NLRP3 inflammasome has been implicated in the pathogenesis of a wide variety of diseases, including Alzheimer's disease, atherosclerosis, and type 2 diabetes[44,45]. Here we additionally established a critical role of Nlrp3 inflammasome in colitis [an IBD that affects millions of people worldwide[46]] consistent with previous findings with Crohn's disease[47,48]. Current therapies for Nlrp3-dependent diseases aim to inhibit the final products of NLRP3 inflammasome (IL-1β and IL-18) or to target inflammasome components[49]. As stated by Pourcet et al., targeting the Rev-erbα/Nlrp3 axis for management of inflammatory diseases is advantageous in its pleiotropic effects, including Ccl2 suppression[11], macrophage infiltration, TLR4 regulation[36], and inactivation of NF-κB signaling (Fig. 6). Another merit of this therapeutic target axis refers to the high drugability of Rev-erbα whose functions can be readily modulated by small-molecule agonists or antagonists[12,43].

It was noteworthy that SR9009 was dosed at ZT8 in animal efficacy studies. This dosing time was chosen as being in coincidence with the highest expression of Rev-erbα protein in normal mice (Supplementary Figure 10A). However, circadian timing system may be altered in mice with DSS-induced colitis as suggested by dysregulated clock genes (Fig. 1). In particular, Rev-erbα expression was downregulated and its rhythmicity was significantly blunted (Fig. 1d). It was therefore acknowledged that ZT8 perhaps was not the optimal dosing time for maximized efficacy in the disease model. Nevertheless, significant pharmacological effects elicited by SR9009 were sufficient to clarify the role of Rev-erbα in colitis development.

In summary, Rev-erbα servers as an integrator of colon clockwork and experimental colitis. Activation of Rev-erbα prevents DSS-induced colitis in mice through its repressive actions on NF-κB and Nlrp3 inflammasome. Targeting Rev-erbα may represent a promising approach for prevention and management of colitis.

## Methods

**Materials.** LPS, ATP, GSK4112, hemin, and thioglycollate broth were purchased from Sigma-Aldrich (St. Louis, MO). Macrophage colony-stimulating factor (M-CSF) was purchased from Peprotech (Rocky Hill, NJ). DSS (molecular weight 36–50 kDa) was obtained from MP Biomedicals (Irvine, CA). SR9009 was purchased from MCE (Monmouth Junction, NJ). Bay11-7082, chemiluminescent EMSA kit and biotin-labeled NF-κB probe were purchased from Beyotime (Shanghai, China). Lipo3000 reagent was purchased from Invitrogen (Carlsbad, CA). ChIP kit was purchased from Cell Signaling Technology (Beverly, MA). RNAiso Plus reagent and PrimeScript RT Master Mix were purchased from Takara (Shiga, Japan). Dual-Luciferase® Reporter Assay system was purchased Promega (Madison, WI). IL-1β, tumor necrosis factor alpha (TNFα), and interferon-γ enzyme-linked immunosorbent assay (ELISA) kits were purchased from Mlbio (Shanghai, China). IL-18, IL-1α, and IL-6 ELISA kits were purchased from Meimian Biotechnology (Yancheng, Jiangsu, China). MPO kit was purchased from Jiancheng Institute of Biotechnology (Nanjing, Jiangsu, China). Murine Raw264.7 cells were purchased from American Type Culture Collection (Manassas, VA).

Antibodies for western blotting are as follows: anti-Rev-erbα (WH0009572M2, Sigma-Aldrich, MO); anti-Nlrp3 (NBP2-12446, CO); anti-IL-1β (AF-401, R&D systems, MN); anti-pro-caspase-1 (14F468, Santa Cruz, CA); anti-caspase-1 p20 (22915-1-AP, Proteintech, Wuhan, China); anti-p65 (10745-1-AP, Proteintech, Wuhan, China); anti-p-p65 (#3031, CST, MA); anti-p-IKBα (14D4, CST, MA); anti-Histone H3 (17168-1-AP, Proteintech, Wuhan, China); and anti-β-actin (ab8226, Abcam, Cambridge, UK). For ChIP assays, antibody against Rev-erbα and normal rabbit IgG were purchased from Cell Signaling Technology (Beverly, MA). For immunofluorescence analysis, Alexa Fluor 488-conjugated anti-mouse antibody was purchased from Life Technologies (Gaithersburg, MD).

*Plasmids:* pGL4.11 and pRL-TK vectors were purchased from Promega (Madison, WI). *Bmal1* (2 kb)-Luc, *Nlrp3* (2000)-Luc, *Nlrp3* (1310)-Luc, *Nlrp3* (1100)-Luc, *Nlrp3* (800)-Luc, *p65* (2000)-Luc, *p65* (1200)-Luc, *p65* (800)-Luc, pcDNA-Rev-erbα, and pcDNA-Rev-erbβ were synthesized by Biowit Technologies (Shenzhen, China). Small interfering RNA targeting Rev-erbα was purchased from Transheep (Shanghai, China). Luciferase reporter constructs containing mutated versions of RevRE sites were obtained from Biowit Technologies (Shenzhen, China). NF-κB-dependent reporter (containing four κB sites) was purchased from Beyotime (Shanghai, China).

**Animals.** Wide-type C57BL/6 mice were obtained from Beijing HFK Bioscience (Beijing, China). All genetic mice were created on a C57BL/6 background. *Bmal1*$^{-/-}$ mice were generated using the CRISPR/Cas9 system (Bioray Laboratory, Shanghai, China). *Rev-erbα*$^{-/-}$ mice were generated using the CRISPR/Cas9 system (Cyagen Biosciences Inc., Guangzhou, China). *Nlrp3*$^{-/-}$ mice (B6.129S6-Nlrp3$^{tm1Bhk}$/J) were obtained from Jackson Laboratory. All mice were bred and housed in Institute of Laboratory Animal Science (Jinan University, Guangzhou, China). Eight- to fourteen-week-old mice (male) were used for in vivo experiments. All animal care and experimental procedures were in compliance with guidelines approved by the Institute of Laboratory Animal Science of Jinan University (Guangzhou, China). Mice were randomly allocated into experimental groups based on body weight. Sample size was determined according to preliminary experimental observations. No data were excluded.

**DSS-induced colitis model.** Acute colitis was induced by feeding mice with 2.5% (w/v) DSS (dissolved in drinking water) for 8 days. Mice were sacrificed on day 8, and colons were collected for biochemical analyses. The colon length was measured with a centimeter ruler. To evaluate the effects of SR9009 on colitis, SR9009 (50 mg/kg) was administered to mice via intraperitoneal injection once daily at ZT8 (corresponding to a peak expression of Rev-erbα, Supplementary Figure 10A) for 7 days prior to DSS treatment, and SR9009 dosing was continued along with DSS treatment (Supplementary Figure 10B). Colon tissues were fixed in 4% paraformaldehyde and embedded in paraffin, followed by hematoxylin-eosin staining. Histological damage was scored based on goblet cells loss, mucosa thickening, inflammatory cells infiltration, submucosa cell infiltration, ulcers, and crypt abscesses. A score of 1–3 or 1–4 were given for each parameter (scoring criteria provided in Supplementary Table 6) with a maximal total score of 20. DAI scores were determined based on body weight loss, occult blood, and stool consistency. A score of 1–4 was given for each parameter with a maximal total score of 12.

**Isolation of primary macrophages.** Mice were injected intraperitoneally with 4% thioglycollate broth. Four days later, peritoneal fluid was collected and plated in 1640 supplemented with 10% fetal bovine serum (FBS). Two hours later, nonadherent cells were aspirated and adherent cells (PMs) were obtained. BMDMs were differentiated from tibial and femoral bone marrow aspirates of mice. Cells were cultured with 1640 supplemented with 10% FBS and 20 ng/ml recombinant murine M-CSF. Seven days later, adherent macrophages were obtained and plated in 12-well plates.

**Quantitative polymerase chain reaction.** Total RNA was isolated using RNAiso Plus reagent and reverse-transcribed using the PrimeScript RT Master Mix. PCR amplification procedure consists of an initial denaturation at 95 ℃ for 5 min, followed by 40 cycles of denaturation at 95 ℃ for 15 s, annealing at 60 ℃ for 30 s, and extension at 72 ℃ for 30 s. 18s RNA was used as an internal control. Relative expression was calculated by using the $2^{-\Delta\Delta CT}$ method. Primer sequences are provided in Supplementary Table 1.

**Luciferase reporter assay.** Cells were transfected with 500 ng of luciferase reporter plasmids, 50 ng of pRL-TK vector (an internal control with renilla luciferase gene), and 500 ng expression plasmids (Rev-erbα or Rev-erbβ). The transfection assays were performed using Lipo3000 according to the manufacturer's protocol. On the next day, the medium was changed to phenol-free Dulbecco's modified Eagle medium with or without SR9009. Luciferase activities were determined by the Dual-Luciferase® Reporter Assay System and GloMax™ 20/20 luminometer (Promega). The relative luciferase activity values of treated cells were normalized to that of control cells.

**EMSA and ChIP.** For EMSA assays, the nuclear proteins were prepared using a cytoplasmic/nuclear protein extraction kit. The DNA–protein complex was loaded onto 4% nondenaturing polyacrylamide gels. After 35-min electrophoresis in 0.25× Trisborate-EDTA buffer, the products in the gels were transferred to Hybond-N$^+$

membranes. The signals were visualized by Omega Lum G imaging system (Aplegen, CA). Oligonucleotides sequences are provided in Supplementary Table 2.

ChIP assays were performed using the Enzymatic Chromatin IP Kit (Magnetic Beads). PMs were cultured in 10-cm dishes and treated with SR9009 (10 μM) or vehicle for 1 h, followed by addition of LPS. Cells were then crosslinked with 1% formaldehyde for 20 min at room temperature. After termination by the addition of glycine, DNA was digested with Micrococcal Nuclease and sheared chromatin was immunoprecipitated with anti-Nr1d1 or normal rabbit IgG. Immunoprecipitated chromatin was decrosslinked at 65 °C for 4 h and purified by using spin columns. Primer sequences are provided in Supplementary Tables 2 and 3.

**Western blotting**. The protein samples were subjected to sodium dodecyl sulfate-polyacrylamide gel electrophoresis (10% acrylamide gels). The resulting products were transferred to polyvinylidene difluoride membranes, followed by blocking with 5% skimmed milk. The membranes were incubated with primary antibodies (1:3000 dilution for anti-β-actin and anti-Histone H3, and 1:1000 dilution for all other antibodies), followed by incubation with anti-mouse (1:2000 dilution), anti-rabbit (1:5000 dilution), or anti-goat (1:5000 dilution) horseradish peroxidase-conjugated secondary antibody. The blots were visualized with enhanced chemi-luminescence and imaged by Omega Lum G imaging system (Aplegen). Uncropped scans of representative blots are provided in Supplementary Figure 13. Protein bands were quantified with densitometry using Quantity One software (Bio-Rad).

**RNA-sequencing**. Nine normal mice and nine mice with colitis were sacrificed at each circadian time point (ZT0, ZT8, and ZT16), and the colons were isolated, snap-frozen, and stored at −80 °C. Total RNA was extracted using RNAiso Plus reagent and RNA quality was analyzed by 2100 BioAnalyzer Expert (Agilent). Three RNA pools (i.e., experimental samples) for normal and colitis groups (at each circadian time point) were assembled by mixing equal amounts of RNA from three animals. Each of three pools was subjected to RNA-sequencing. In brief, an aliquot of RNA pool (about 1 μg) was used for construction of NEB libraries with NEBNext Ultra RNA Library Prep Kit for Illumina (NEB, Ispawich, MA, USA). The library preparations were sequenced on an Illumina Hiseq X TEN platform (Novogene, Beijing, China), generating 150 bp paired-end reads. Reads were aligned to the hg38 genome using Maximal Mappable Prefix method by STAR (v2.5.1b). Fragments per kilobase of transcript per million mapped reads were calculated using RSEM software (v1.2.28). DEGs were obtained by comparing gene expressions between two groups using DESeq2 (v1.10.1). Corrected $P$ value of <0.05 was considered statistically significant. Heatmaps were generated by R package with hierarchical clustering algorithm.

**Immunofluorescence analysis**. RAW264.7 cells were pretreated with 10 μM SR9009 for 1 h and stimulated with 100 ng/ml LPS for 8 h. Cells were fixed, permeated, blocked, and incubated with primary antibody against NF-κB phospho (p)-p65 overnight at 4 °C. Cells were probed with an Alexa Fluor 488-conjugated anti-mouse antibody (1:1000) for 1 h at 37 °C. Then, 1 ml 4′,6-diamidino-2-phenylindole (Biotium) was added to the culture flask for 20 min, followed by washing with phosphate-buffered saline. Fluorescent images were collected by laser scanning microscope (Carl Zeiss, Oberkochen, Germany).

**Statistical analyses**. Data are recorded as mean ± SD. Statistical differences between two groups were analyzed by Student's $t$ test or non-parametric tests (Mann–Whitney $U$ test) as appropriate. Statistical analysis for survival curve was performed with the log-rank test. The level of significance was set at $*P < 0.05$.

## Data availability
RNA-seq data have been deposited to Sequence Read Archive (SRA) with a SRA accession of SRP149957. Other data are available upon request.

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

## Acknowledgements
This work was supported by the National Natural Science Foundation of China (No. 81722049 and 81573488).

## Author contributions
B.W. and S.W. designed the study; S.W., Y.L., X.Y., F.L. and L.G. performed experiments; S.W., Y.L. and X.Y. collected and analyzed data; B.W., S.W. and Y.L. wrote the manuscript. S.W. and Y.L. contributed equally to this work.

## Additional information

**Competing interests:** The authors declare no competing interests.

