## [Peer Review File · Nature Communications]

Reviewers' comments:

Reviewer #1 (Remarks to the Author):

The manuscript titled REV-ERBa integrates colon clock with experimental colitis through regulation of NFkB/NLRP3 axis is an interesting, well written manuscript. The authors have elegantly described a mechanism of how gut clock –circadian rhythm might be involved in the development of intestinal inflammation. The authors have utilised all the relevant methodology to nail the mechanism of Nf-kB-NLRP3 axis in the gut-circadian rhythm connection through REV-ERBa. While I do think the authors have convincingly show the connection, I have the following questions/comments;

Well written introduction but, too short on the actual connection between circadian rhythm and colitis development. Also, the recent link with gut microbiota is unaddressed.

The methods employed are appropriate but I am a bit concerned that the whole of inflammatory connection is the use of just the DSS model for colitis. Another point I want to verify is that the authors have pre-treated with SR9009 for 7 days to show how REV-ERBa activation reduced colitis. What is the possible explanation for the effect here? The promoter methods are all appropriate and well conducted.

Was there any gender-specific defect seen in these experiments? Recent literature suggests that gut microbiota changes are regulated during the circadian clock disturbance. I am sure the authors have data on microbiota changes. Can they provide more on that aspect? Also, can the authors describe the distribution of clock genes in the entire digestive tract (small intestine, cecum, ileum, jejunum, proximal and distal colon)?

While the IL-1b levels are well described in the paper, IL-18 was initially shown as regulated along with IL-1b, the authors mainly sowed only IL-1b for the rest of the paper. Can authors comment if IL-18 is regulated along with IL-1b by REV-ERBa? Specifically, comment if both the NLRP3 mediated cytokines are important for colitis development in their models?

In a translational sense, how would treatments for jet lag such as melatonin would work? Can the authors comment on how their findings correlate with human jet lag conditions? Would the authors also comment on any other pathologies seen in clock deficit mice? One more aspect I would like to know about is the effect with respect the described mechanism in aging? Do the authors have any data in aged mice?

In Fig 2, can the authors provide more images of intestinal histopathoogy? In Fig 2K, why did the authors only use peritoneal macrophages and why not bone-marrow derived macrophages? I am not completely convinced that Rev-erba is a gatekeeper of intestinal inflammation as claimed by the authors. Need to substantiate with a lot more gut pathology data. Also, IL-1b is not the only cytokine that causes colitis. There are other cytokines such as TNF. I also wonder if there are compensatory up/down regulation of other inflammasomes while Nlrp3 is blocked?

Reviewer #2 (Remarks to the Author):

The relevance of the molecular clock for the development of ulcerative colitis is an important topics, which is convincingly addressed in results 2.1-2.4, with minor concerns. However, there were several methodologic concerns in the subsequent sections.

- Three timepoints are insufficient for properly assessing circadian rhythmicity. This is especially true within the context of a circadian disruption where phase shifting could indeed be taken for severe dampening, if peaks and troughs are missed or the pattern is distorted rather than suppressed, due to an insufficient sampling frequency.

- There is no mention of the several pharmacologic properties ascribed to this molecule, one of which being that of a REV-ERB agonist (Li & Laher, TIPS 2015). While SR9009 convincingly prevented DSS-induced colitis through reducing Nirp3 and IL1beta mRNAs and proteins in Rev-erb - proficient mice, no such data are provided in Rev-erb deficient mice, despite they displayed most severe DSS-ulcerative colitis. Does the alleged Rev-erb agonism of SR9009 require Rev-erb

expression? Interestingly SR9009 dosing at ZT0 did not alter the circadian pattern in Rev-erb (Amador et al PLOS One 2016). Moreover, it is not explained why SR9009 had no effect following dosing after DSS-colitis had been induced.

- The several hundred-fold changes in Rev-erb mRNA expression from a low point near ZT15-18 to a high point near ZT3-6 in mouse liver or colon (Li et al. Cancer Res 2013) are not considered in many comparisons that involve a single timepoint in the manuscript. there are many missing information in the Methods (mouse sex and age, synchronisation and jet lag methods, method of measuring colon length (or thickness?), dosing time of SR9009 (which could matter).

Reviewer #3 (Remarks to the Author):

In this manuscript the authors investigate the potential role of the circadian clock protein Rev-erba in colitis using the DSS mouse model. Inflammatory bowel disease has been linked to perturbation of the circadian timing system yet little is known about the underlying mechanisms. In that respect, the study is addressing an important and original question with a translational potential. This is a quite detailed study with conclusions that are fairly well supported by the reported data. The study reveals a new role for Rev-erba which appears to be a pivotal protein at the crossroad between circadian timing, metabolism and inflammation. However there is a considerable lack of rigor and information throughout the paper that cast doubt about the ms in general.

Major concerns:

- All western blot analyses show oversaturated loading controls (b-actin) and although not indicated, it seems to me that some proteins shown in the same composite figures may have been analyzed using different membranes and/or probing. It is therefore very difficult to understand how the authors have rigorously evaluated the expression level of the proteins of interest. This is a major issue that needs to be addressed. Expression levels must be properly quantified and subjected to statistical testing. Indeed, western blots show only one replicate. It is critical that the authors show that they have replicated these analyses and got statistically validated differences consistent with their claims. In addition, many western blots are misaligned and cut sharp. Unedited western blots must be provided in the supplemental data.

- In fig 1, the authors report on a RNAseq experiment but they do not provide any detail about how the experiment was done (replicates, parameter values used for the data processing, etc). The Panther analysis shown in fig 1B should be extended to other categories to provide a more global picture of the DSS effect.

- In fig 2 the authors used two genetic models, Bmal1^{-/-} and Rev-erba^{-/-}. These models are not the ones used in previous published studies and were generated using a different technology (crisper/cas 9). It is absolutely critical that the authors validate these models with regard to clock dysfunction. A wheel running activity recording and clock gene expression profiling in the periphery should be provided. The lack of the BMAL1 or REV-ERBa protein in these two KO needs also to be demonstrated.

- In fig 5, the authors report a mechanism for the repression of Nlrp3 by Reverba. This predicts that NlrP3 is also a target of the RORa/g nuclear receptor. Activation of NlrP3 by RORa/g should be tested as well as the antagonism between Rev-erba and RORa/g. The same concern applies to p65.

- The experiment in fig 6E should be quantified as a conclusion cannot be drawn based on a couple of cells. In fig 6F, the result is not convincing as the b-actin level in the nr1d1 condition seems higher, so is there really a decreased of p65 expression. Why are the b-actin bands not centered with the p65 signal, this gives the impression that both proteins were not from the same membrane.

- In the method section, the authors very often refer to previous publications without giving any single detail. More information is required including protocol modifications, otherwise the section is useless!

- The authors use the t-test for comparisons between genotypes/treatments. This is not the

adequate method as the number of replicates is often small and normality cannot be proven (n =5). Non-parametric tests are required.

Others concerns

- Actograms should be shown as doubleplots and one should see the transition from LD to JL for the exact same representative mice. Further a statistical analysis should also be provided for these measurements for a sufficient number of mice.
- What is the circadian time used in the comparison shown in fig 2A ?
- In the result section, the 2.2 and 2.3 may be grouped together as there some redundancy between the two paragraphs.
- In fig 5 E, I don't see a clear effect of the competitor oligo. EMSA is not quantitative, outdated technique and, of no use since a Chip experiment is shown. The same comment applies to fig 6H.
- The sex of the animals used is not indicated, were they males, females or both ?
- In p4 (top) of the introduction, authors should refer to Rev-erb a and b.
- Fig 6B-D does not include confocal microscopy as indicated in the text.

We wish to thank the editor and reviewers for careful and valuable reviews. Below are our point-by-point responses.

Responses to Editor

“Currently the circadian analyses are too narrow requiring expansion (reviewer 2).”

Response: We accept the editor’s criticism. Accordingly, we have performed new experiments [i.e., qPCR analyses of circadian expression of core clock genes (Figure 1D), circadian expression of Rev-erba in colons (Figure 2B), and circadian regulation of Nlrp3 and its related genes (Figure 3B)] wherein six circadian time points were sampled and analyzed. The relevant data have been updated (please see new Figure 1D, Figure 2B & Figure 3B in revised version).

“Extension of the SR9009 experiments (reviewer2) is also required to strengthen the support for your conclusions.”

Response: We have extended the SR9009 experiments to Rev-erba deficient mice in revised version (Supplementary Figure 10). SR9009 failed to alleviate DSS-induced colitis in Rev-erba-deficient mice, supporting requirement of Rev-erba expression for SR9009 action on colitis (Supplemental Figure 10). Also we have performed new SR9009 experiments to determine the potential benefits of Rev-erba activation on management of colitis in addition to colitis prevention (Supplementary Figure 11C). SR9009 administration post DSS challenge showed moderate protective effects on colitis development (Supplemental Figure 11C).

“Additionally, the requirement and role of Rev-erb alpha in colitis prevention requires further exploration (reviewers 1 and 3).”

Response: Following the editor and reviewer’s suggestions, multiple sets of new experiments concerning the role of Rev-erba in colitis prevention have been performed, and the new data have been added and described in revised version. In particular, we have performed new experiments to confirm the role of Rev-erba in regulation of Nlrp3 inflammasome using bone marrow derived macrophages (Supplementary Figure 6). The role of IL-18 in colitis prevention by Rev-erba activation was also pinpointed (Supplementary Figure 6). In addition, we showed Rora (playing an antagonistic role to Rev-erba in circadian regulation) activated the expressions of Nlrp3/p65, and attenuated the repressor actions of Rev-erba on Nlrp3 /p65 consistent with the repressive effects of Rev-erba on Nlrp3/p65 (Supplementary Figure 9).

“Reviewer 3 also suggests validation of the novel in vivo models which we also agree will be of import to support the associated data from these animal models.”

Response: We have followed the reviewer’s suggestion to further validate the genetic models. Accordingly, we have provided the data of wheel running activity, gene expression profiling, and protein expression. The new data have been added to Figure 3B and Supplementary Figures 3 & 4.

Responses to reviewer #1

The manuscript titled REV-ERNa integrates colon clock with experimental colitis

through regulation of NFkB/NLRP3 axis is an interesting, well written manuscript. The authors have elegantly described a mechanism of how gut clock –circadian rhythm might be involved in the development of intestinal inflammation. The authors have utilised all the relevant methodology to nail the mechanism of Nf-kB-NLRP3 axis in the gut-circadian rhythm connection through REV-ERBa. While I do think the authors have convincingly show the connection, I have the following questions/comments;

Well written introduction but, too short on the actual connection between circadian rhythm and colitis development. Also, the recent link with gut microbiota is unaddressed.

Response: The reviewer raised a comment that more information should be provided in INTRODUCTION regarding the actual connection between circadian rhythm and colitis development. Previous studies reported that disruption of circadian rhythm significantly worsens the development of colitis (Am J Physiol Regul Integr Comp Physiol. 2008;295(6):R2034-40 // Sleep Med. 2009;10(6):597-603), indicating an interconnection between circadian rhythms and colitis. However, the mechanism for this connection is still unknown. Therefore, it is stated in the INTRODUCTION (the third paragraph) that “Disruption of circadian rhythms is reported to increase the risks for developing IBD.” We are unable to expand this point due to the lack of mechanistic studies.

The reviewer raised a second comment that the authors need to include the link of circadian rhythm with gut microbiota. We agree with the reviewer. Accordingly, we have added the following sentence to the INTRODUCTION (the third paragraph).

“Circadian perturbation also has the potential to alter gut microbiota, contributing to IBD pathogenesis²¹⁻²³”

The methods employed are appropriate but I am a bit concerned that the whole of inflammatory connection is the use of just the DSS model for colitis.

Response: The reviewer is concerned about only DSS model for colitis. Animal models of IBD are valuable and indispensable tools for investigating the pathogenesis of IBD and for evaluating different therapeutic options. Various animal models of IBD (including the models induced by a chemical irritant or bacterial infection and transgenic or gene knockout mouse strains) have been documented in the literature (Drug Des Devel Ther. 2013;7:1341-57). However, DSS-induced colitis model has some advantages when compared to other animal models of colitis. It is well appreciated and widely used because of simplicity, reproducibility and controllability as well as many similarities with human ulcerative colitis (J Biomed Biotechnol. 2012;2012:718617 // Curr Protoc Immunol. 2014;104:Unit 15.25). In fact, DSS-induced colitis can be used as a relevant model for the translation of mice data to human diseases (Int Immunopharmacol. 2008;8(6):836-44).

As stated by Low *et al.* (Drug Des Devel Ther. 2013;7:1341-57), IBD models should be carefully chosen according to the particular question to be addressed. Our study aims to address the underlying mechanisms for circadian regulation of colonic inflammation. DSS model mimics well the aberrant (innate) immune responses in UC. Also, DSS model has been applied to the evaluation of NLRP3 inflammasome responses in a series of studies (Cell Mol Gastroenterol Hepatol. 2015;1(2):154-170). Therefore, we believe that the DSS model is appropriate and sufficient to establish the potential link between circadian clock and colitis, and to investigate the underlying mechanisms.

As discussed above, DDS model is particularly useful to study the innate immune mechanisms in intestinal inflammation (an acute inflammatory response independent of T and B cells) (Gut. 2010;59(9):1192-9). In fact, many excellent studies investigate the pathogenesis of colitis based on this model only. For example, Neudecker *et al.*

reported that miR-223 plays an important role in regulating the innate immune response during intestinal inflammation (J Exp Med. 2017;214(6):1737-1752). Zhou et al. revealed a significance role of intestinal PPAR-UGT and FXR-FGF15 signaling in the pathological development of colitis (Nat Commun. 2014;5:4573) by using DSS-colitis model. Therefore, it should be of little concern that DDS model was used herein to uncover the circadian regulation of colitis.

Another point I want to verify is that the authors have pre-treated with SR9009 for 7 days to show how REV-ERBa activation reduced colitis. What is the possible explanation for the effect here?

Response: It appears that the reviewer is questioning why pre-treatment of SR9009 was performed. The purpose of pre-treatment to ensure the steady-state concentrations (highest drug exposures, based on pharmacokinetic theory) and to maximize activation effects of Rev-erba prior to colitis induction. This may be necessary because SR9009 is concerned with poor pharmacokinetic properties (i.e., short half-live and high clearance) (J Med Chem. 2013;56(11):4729-37). We provided strong evidence in current study that Rev-erba activation alleviates experimental colitis by down-regulating the NF- κ B /NLRP3 axis. To be specific, REV-ERBa activation both directly and indirectly (via p65) repressed Nlrp3 transcription and decreased activation and release of IL-1 β and IL-18 to alleviate the colitis.

The promoter emthods are all appropriate and well conducted. Was there any gender-specific defect seen in these experiments?

Response: We are unable to answer the reviewer's question because gender-comparison studies were not conducted (all *in vivo* studies were based on male mice). Such comparison studies are NOT justified because of NO gender difference in risks of developing UC (Gastroenterol Clin North Am. 2002;31(1):1-20). IBD generally occurs with equal frequency in men and women (Clin Gastroenterol Hepatol. 2007;5(12):1424–1429).

Recent literature suggests that gut microbiota changes are regulated during the circadian clock disturbance. I am sure the authors have data on microbiota changes. Can they provide more on that aspect?

Response: Thanks very much for the information. The reviewer is right that previous report shows that circadian disruption causes changes in gut microbiota although the underlying mechanisms remain unaddressed (Note that this information has been added to the INTRODUCTION section) (PLoS One. 2014;9(5):e97500). Current study focuses on circadian regulation of colonic inflammation by REV-ERBa. We did not intend to study circadian regulation of gut microbiota or the role of microbiota in intestinal inflammation, thus we don't have data on microbiota changes. We believe that investigating a potential role of microbiota in intestinal inflammation should be an independent work.

Also, can the authors describe the distribution of clock genes in the entire digestive tract (small intestine, cecum, ileum, jejunum, proximal and distal colon)?

Response: Distribution of clock genes (e.g., Clock, Bmal1, Per, Cry, Rev-erba) in murine gastrointestinal track has been well established in the literature (Gastroenterology.2007;133(4):1250-60// J Lipid Res. 2009;50(9):1800-13// Chronobiol Int. 2009;26(4):607-20). Clock genes are expressed in all regions of the gut, including the stomach, all parts of small intestine and colon (Gastroenterology.2007;133(4):1250-60// J Lipid Res. 2009;50(9):1800-13). Since the information is available in the literature, it may be unnecessary to describe again in current manuscript the intestinal distribution of clock genes.

While the IL-1b levels are well described in the paper, IL-18 was initially shown as regulated along with IL-1b, the authors mainly sowed only IL-1b for the rest of the paper. Can authors comment if IL-18 is regulated along with IL-1b by REV-ERBa? Specifically, comment if both the NLRP3 mediated cytokines are important for colitis development in their models?

Response: The reviewer raised a good question whether IL-18 is regulated by REV-ERBa. In fact, in original submission we have shown that REV-ERBa represses the transcription of IL-18. To be specific, IL-18 showed similar changes along with IL-1 β upon Rev-erba overexpression, Rev-erba knockdown and activation by SR9009 (Supplemental Figure 6 in original submission). Hence, REV-ERBa does regulate IL-18. Rev-erba regulation of IL-18 (and IL-1 β) is attained through down-regulation of NF- κ B that is a transcriptional activator of IL-18 (and IL-1 β) (PLoS Pathog. 2015;11(7):e1004948).

We regret that fewer IL-18 data were provided in the original submission. In revised version, we have added some data related to IL-18 changes in response to Rev-erba activity manipulation. First, Rev-erba deletion caused a reduction in IL-18 protein along with IL-1b (ELISA experiments, please see new Figure 2J). Second,

Rev-erba activation decreased expression of mature IL-18 protein along with mature IL-1 β (please see new Figure 4G and Supplemental Figure 6B).

The reviewer raised a second question if both the NLRP3 mediated cytokines are important for colitis development. It seems that the reviewer is concerned about the role of IL-18 in colitis due to relatively insufficient data of IL-18 characterization. To alleviate the reviewer's concern, we have performed new experiments for IL-18 measurements. We found that both mRNA and protein expression of IL-18 showed similar changes with IL-1 β (new Figure 7H,I &J) consistent with the in vitro results (new Figure 4G and Supplemental Figure 6A&B). Decreased IL-1 β and IL-18 in alleviated DSS-induced colitis clearly indicates that both NLRP3 mediated cytokines are important for colitis development.

In a translational sense, how would treatments for jet lag such as melatonin would work?

Response: The reviewer raised an interesting question how treatments for jet lag such as melatonin would work? We are unable to provide precise answers to this question because no relevant experiments were performed. However, it is interesting to note that melatonin has potential to improve experimental colitis (Inflamm Bowel Dis. 2009;15(1):134-40 // Int J Mol Med. 2015;35(4):979-86). Although the exact mechanisms remain unknown, this effect may result from melatonin's inhibition and/or suppression of specific inflammation-related cytokines and cell adhesion molecules (Inflamm Bowel Dis. 2009;15(1):134-40). Current study identifies REV-ERBa as an integrator of circadian rhythms and experimental colitis, highlighting REV-ERBa/NLRP3 axis as a therapeutic target for colitis management. Since melatonin is not a REV-ERBa modifier, melatonin modulation of colitis via the REV-ERBa pathway is unlikely.

Can the authors comment on how their findings correlate with human jet lag conditions?

Response: The reviewer asked for a comment on how the findings correlate with human jet lag conditions. However, such comment may be premature due to the lack of relevant human data. Current study aims to address the underlying mechanisms for circadian regulation of IBD diseases. Although it is not completely certain that all novel findings based on mice can be translated into humans, disruption of circadian rhythms (including jet lag) increases the risks of colitis development in both mice and humans (Expert Rev Clin Immunol. 2011;7(1):29-36 // Pharmacol Rep. 2016;68(4):847-51), and REV-ERBa is a functionally conserved protein across species (J Mol Endocrinol. 2004;33(3):585-608 // J Lipid Res. 2002;43(12):2172-9). Therefore, there is a high possibility that REV-ERBa regulation of colitis also applies to humans.

Would the authors also comment on any other pathologies seen in clock deficit mice?

Response: The pathologies caused by genetic deletion of Bmal1 or Rev-erba (i.e., the potential physiological functions of the clock genes) have been well documented in the literature (PLoS Biol. 2004;2(11):e377 // Hum Reprod Update. 2005;11(1):91-101 // Genes Dev. 2006;20(14):1868-73 // Cell 2002;110(2):251-60). For instance, in addition to impaired circadian behavior, mice deficient in Bmal1 display an accelerated aging phenotype associated with non-orthotropic ossifications and lower bone weights (Genesis. 2005;41(3):122-32 // Genes Dev. 2006;20(14):1868-73). Loss of Rev-erba results in disrupted circadian behavior and

metabolic shift (e.g., altered glucose and lipid profiles) (Nature. 2012;485(7396):123-7 // Genes Dev 2012;26(7):657-67). Since current study aimed to investigate circadian regulation of colonic inflammation, it was reasonable to center our work on the inflammatory dissection other than other biological processes. We believe that it is unnecessary to comment on other pathologies caused by deletion of clock genes.

One more aspect I would like to know about is the effect with respect the described mechanism in aging? Do the authors have any data in aged mice?

Response: The reviewer is curious about whether the mechanisms described are applicable to aged mice. We believe that this question is beyond the scope of current manuscript.

In Fig 2, can the authors provide more images of intestinal histopathoogy?

Response: Revised as suggested.

In Fig 2K, why did the authors only use peritoneal macrophages and why not bone-marrow derived macrophages?

Response: Peritoneal macrophages and bone-marrow derived macrophages are two commonly used in vitro macrophage models in biomedical research. There is no

particular reason as to why we have to use peritoneal macrophages. To alleviate the reviewer's concern. We have performed new experiments using bone-marrow derived macrophages. Rev-erba activation similarly suppressed Nlrp3 inflammasome activity in bone-marrow derived macrophages (new Supplemental Figure 6C).

I am not completely convinced that Rev-erba is a gatekeeper of intestinal inflammation as claimed by the authors. Need to substantiate with a lot more gut pathology data.

Response: To alleviate the reviewer's concern, we have softened our statement. In revised manuscript, this statement has been changed to "Our data suggest Rev-erba as a potential gatekeeper of intestinal inflammation."

Also, IL-1b is not the only cytokine that causes colitis. There are other cytokines such as TNF. I also wonder if there are compensatory up/down regulation of other inflammasomes while Nlrp3 is blocked?

Response: The reviewer is curious about the role of TNF in colitis development. In fact, we have shown that TNFa was slightly suppressed by Rev-erba (Supplemental Figure 6D). This effect most likely was attained via Rev-erba down-regulation of NF- κ B which is a transcriptional activator of TNFa (J Clin Invest. 2001;107(1):7-11 // PLoS One.2010;5(3):e9585). However, we believe that IL-1B and IL-18 play a dominant role in colitis alleviation by Rev-erba activation because Rev-erba activation directly repress Nlrp3 expression and activity (inhibiting maturation of IL-1B and IL-18

proteins), and the protective effect of SR9009 on colitis was lost upon Nlrp3 deletion (Figure 7).

The reviewer is also curious about whether there is compensatory regulation of other inflammasomes while Nlrp3 is deleted. We believe that the compensation, if there is any, would be negligible. This is because the protective effect of SR9009 on DSS challenge ceases to exist in Nlrp3^{-/-} mice (Figure 7).

Responses to reviewer #2

The relevance of the molecular clock for the development of ulcerative colitis is an important topic, which is convincingly addressed in results 2.1-2.4, with minor concerns. However, there were several methodologic concerns in the subsequent sections.

- Three timepoints are insufficient for properly assessing circadian rhythmicity. This is especially true within the context of a circadian disruption where phase shifting could indeed be taken for severe dampening, if peaks and troughs are missed or the pattern is distorted rather than suppressed, due to an insufficient sampling frequency.

Response: We accept the reviewer's criticism that three time points were insufficient for proper assessment of circadian rhythmicity. Accordingly, we have performed new experiments [qPCR analyses of circadian expression of core clock genes (Figure 1D), circadian expression of Rev-erba in colons (Figure 2B), and circadian regulation of Nlrp3 and its related genes (Figure 3B)] wherein six circadian time points were sampled and analyzed. The relevant data have been updated (please see new Figure 1D, Figure 2B & Figure 3B in revised version).

- There is no mention of the several pharmacologic properties ascribed to this molecule, one of which being that of a REV-ERB agonist (Li & Laher, TIPS 2015). While SR9009 convincingly prevented DSS-induced colitis through reducing Nirp3 and IL1beta mRNAs and proteins in Rev-erb - proficient mice, no such data are provided in Rev-erb deficient mice, despite they displayed most severe DSS-ulcerative colitis. Does the alleged Rev-erb agonism of SR9009 require Rev-erb expression? Interestingly SR9009 dosing at ZT0 did not alter the circadian pattern in Rev-erb (Amador et al PLOS One 2016). Moreover, it is not explained why SR9009 had no effect following dosing after DSS-colitis had been induced.

Response: The reviewer raised a comment that additional reported biological effects of REV-ERBa (Trends Pharmacol Sci. 2015;36(12):906-917) should be included. The reviewer is right that REV-ERBa has a function that was not mentioned in original submission. That is REV-ERBa enhances mitochondrial biogenesis and improves oxidative function, potentially impacting energy metabolism. The REV-ERBa agonist SR9009 may be a promising exercise pill that mimics exercise-like benefits on energy metabolism (Trends Pharmacol Sci. 2015;36(12):906-917). In the revised manuscript, we have added this biological effect of REV-ERBa in the INTRODUCTION section (the last sentence of the second paragraph).

The reviewer raised a good comment that the authors need to provide the data of SR9009 effect on colitis development in Rev-erba-deficient mice. We agree with the reviewer on this matter. Accordingly, we have performed new SR9009 experiments using Rev-erba-deficient mice. We demonstrate that the protective effect of SR9009 on colitis development was lost in Rev-erba-deficient mice (no changes were observed in Nirp3 and IL1 beta and IL-18 expressions). The new data were added to Supplemental Figure 10. The new results have been described in the RESULTS

section in revised version (under the subheading of “2.7. Rev-erba activation alleviates experimental colitis in mice”).

The reviewer is right that SR9009 dosing did not alter the circadian pattern in Rev-erba (PLoS One. 2016;11(3):e0151014). In fact, we also observed no change in Rev-erba expression following dosing of GSK2945, a Rev-erba antagonist (Drug Metab Dispos. 2018;46(3):248-258). This indicate that the agonist SR9009 enhances Rev-erba activity/function via receptor activation only (recruitment enhancement of the NCoR corepressor complex) (Nature. 2012;485(7396):62-8).

The reviewer asked for an explanation as to why SR9009 had no effect following dosing after DSS-colitis induction. In fact, this is not true. SR9009 administration post DSS challenge also showed moderate protective effects on colitis development (Supplemental Figure 11C), suggesting a dual effect of REV-ERBa activation on prevention and treatment of colitis.

- The several hundred-fold changes in Rev-erb mRNA expression from a low point near ZT15-18 to a high point near ZT3-6 in mouse liver or colon (Li et al. Cancer Res 2013) are not considered in many comparisons that involve a single timepoint in the manuscript. there are many missing information in the Methods (mouse sex and age, synchronisation and jet lag methods, method of measuring colon length (or thickness?), dosing time of SR9009 (which could matter).

Response: The reviewer is right that Rev-erba mRNA showed a robust circadian oscillation (the circadian difference can be several hundred-fold) (Cancer Res. 2013;73(24):7176-88). This is also the case in current study. However, the protein expression was less robust with a maximal difference of ~7-fold (Supplemental Figure 11A). We accept the reviewer’s criticism that data pertaining to circadian genes such

as Rev-erba should be collected at multiple circadian time points for comparisons. Accordingly, for data concerning circadian gene (including Rev-erba) expression, samples at six time points were collected and analyzed (please see new Figure 1D, Figure 2B and Figure 3B). Specifically, for results in Figure 2A, comparisons were reasonably made at both ZT6 (Rev-erba mRNA peak time) and ZT18 (mRNA trough time).

We regret that some information is missing in the Methods. Accordingly, the details of some methods have been added in revised version. In particular, the method for generation of jet lagged mice has been described in Results 2.2 and the dosing time of SR9009 at ZT8 has been specified.

Responses to reviewer #3

In this manuscript the authors investigate the potential role of the circadian clock protein Rev-erba in colitis using the DSS mouse model. Inflammatory bowel disease has been linked to perturbation of the circadian timing system yet little is known about the underlying mechanisms. In that respect, the study is addressing an important and original question with a translational potential. This is a quite detailed study with conclusions that are fairly well supported by the reported data. The study reveals a new role for Rev-erba which appears to be a pivotal protein at the crossroad between circadian timing, metabolism and inflammation. However there is a considerable lack of rigor and information throughout the paper that cast doubt about the ms in general.

Major concerns:

- All western blot analyses show oversaturated loading controls (b-actin) and although not indicated, it seems to me that some proteins shown in the same composite figures may have been analyzed using different membranes and/or probing. It is therefore

very difficult to understand how the authors have rigorously evaluated the expression level of the proteins of interest. This is a major issue that needs to be addressed. Expression levels must be properly quantified and subjected to statistical testing. Indeed, western blots show only one replicate. It is critical that the authors show that they have replicated these analyses and got statistically validated differences consistent with their claims. In addition, many western blots are misaligned and cut sharp. Unedited western blots must be provided in the supplemental data.

Response: The reviewer raised a comment that the loading controls (beta-actin) were “oversaturated” due to much stronger bands. To alleviate the reviewer’s concern, these “oversaturated” beta-actin bands were replaced with the new ones generated with shorter exposure time.

The reviewer is right that some proteins in the same composite figures were analyzed using different membranes. This was reasonable because the same sample (cell or tissue) was often times analyzed multiple times and each time was performed for analysis of different target proteins of interest. We regret that loading controls (beta-actin) were not provided for analysis of every membrane. This has been corrected in revised manuscript (new Figure 4F & 4G).

The reviewer raised another comment that expression levels must be properly quantified and subjected to statistical testing. In fact, all WB experiments were performed three times (three replicates), and the differences between two groups of means were subject to rigorous statistical analyses. We regret that this was not clearly stated in the original submission. In revised manuscript, the relevant information has been included in Figure legends.

We regret that many western blots are misaligned. This has been fixed in revised manuscript.

We believe that it is unnecessary to include unedited Western blot images in Supplemental materials for publication because of extensive number of unedited

western blots. However, we have provided a file containing unedited Western blot images as a separate Supplemental file for your review.

- In fig 1, the authors report on a RNAseq experiment but they do not provide any detail about how the experiment was done (replicates, parameter values used for the data processing, etc). The Panther analysis shown in fig 1B should be extended to other categories to provide a more global picture of the DSS effect.

Response: We regret that some details about RNA-seq experiment were missing in previous submission. Accordingly, more details of RNA-seq have been added to the Methods section (please see the revised manuscript). We agree with the reviewer that pathway analysis should be extended to other categories. Accordingly, the pathway analysis for all categories has been provided in Supplementary Table 4.

- In fig 2 the authors used two genetic models, *Bmal1*^{-/-} and *Rev-erba*^{-/-}. These models are not the ones used in previous published studies and were generated using a different technology (crisper/cas 9). It is absolutely critical that the authors validate these models with regard to clock dysfunction. A wheel running activity recording and clock gene expression profiling in the periphery should be provided. The lack of the BMAL1 or REV-ERBa protein in these two KO needs also to be demonstrated.

Response: The reviewer raised a good comment that the genetic models should be validated. We agree with the reviewer. In fact, we have confirmed the absence of

Bmal1 (or Rev-erba) in the Bmal1 (or Rev-erba) -deficient mice by PCR analyses of mouse tails. To further confirm successful establishment of the genetic models, we have followed the reviewer's suggestion and provided the data of wheel running activity, gene expression profiling, and protein expression. The new data have been added to Figures 3B and Supplementary Figures 3 & 4.

- In fig 5, the authors report a mechanism for the repression of Nlrp3 by Reverba. This predicts that NlrP3 is also a target of the RORa/g nuclear receptor. Activation of NlrP3 by RORa/g should be tested as well as the antagonism between Rev-erba and RORa/g. The same concern applies to p65.

Response: The reviewer raised a good comment that the authors need to test the effects of RORa/g on Nlrp3 because REV-ERB proteins compete with ROR for the same DNA-binding site. We agree with the reviewer and performed new experiments to see if ROR regulates Nlrp3 and P65. Interestingly, RORa/g appeared to activate the transcription of Nlrp3 and p65 genes because Nlrp3 and P65 mRNAs were reduced by SR1001 (an inverse agonist of both ROR α/γ) (Supplementary Figure 9B). Consistently, overexpression of RORa increased Nlrp3 and p65 expressions in Raw264.7 cells (Supplementary Figure 9A). Antagonism between Rev-erba and RORa were also observed (Supplementary Figure 9A). We have described the new data in the RESULTS section (under the subheading of "2.6. Rev-erba represses Nlrp3 transcription via RevRE and NF- κ B binding sites").

- The experiment in fig 6E should be quantified as a conclusion cannot be drawn based on a couple of cells. In fig 6F, the result is not convincing as the b-actin level in

the nr1d1 condition seems higher, so is there really a decreased of p65 expression. Why are the b-actin bands not centered with the p65 signal, this gives the impression that both proteins were not from the same membrane.

Response: The reviewer is right that the quantitative data should be provided for Figure 6E that shows a representative set of images from three separate experiments. Accordingly, the intensity of green fluorescence representing phospho-NF-kB p65 has been quantified in six different fields per view (new Figure 6F).

The reviewer questioned previous Figure 6F data that demonstrate the repression of P65 expression by Rev-erba (Nr1d1). The reviewer seems to make a wrong comparison because an apparently higher beta-actin (as the normalizer) level in the Nr1d1 group indicates a much more reduction of P65 protein than judged from the P65 bands alone.

The reviewer raised a comment why the beta-actin bands are not centered with the P65 signal. This is simply because the signal of beta-actin is much stronger than p65, and it looks like these bands are not centered. To alleviate the reviewer's concern, this experiment was re-performed and the figure has been replaced with a new one in the revised version (new Figure 6G).

- In the method section, the authors very often refer to previous publications without giving any single detail. More information is required including protocol modifications, otherwise the section is useless!

Response: We accept the reviewer's criticism that some details are lacking in the Methods section. Accordingly, more details have been added to the experiment protocols (please see the revised manuscript).

- The authors use the t-test for comparisons between genotypes/treatments. This is not the adequate method as the number of replicates is often small and normality cannot be proven ($n = 5$). Non-parametric tests are required.

Response: Thanks for the suggestion. We have followed the reviewer's suggestion. Normality tests were performed to determine whether data sets are well-modeled by a normal distribution. T-test was performed for mean comparison when normality was met. Non-parametric test (Mann–Whitney U test) has been used for comparisons when normality cannot be met.

Others concerns

- Actograms should be shown as doubleplots and one should see the transition from LD to JL for the exact same representative mice. Further a statistical analysis should also be provided for these measurements for a sufficient number of mice.

Response: We have followed the reviewer's suggestion and the actograms have been shown in the double plot format. It may be not clear to the reviewer that wheel running activity was monitored for mice whose circadian rhythms have been disrupted by executing a jet lag protocol. The purpose of wheel running experiment was to confirm the successful model establishment. Therefore, the so-called transition from LD to JL may be not applicable (please see Cell Metab. 2016;24(2):324-31). Each group of wheel running experiment was performed with six mice ($n = 6$). The average of running activities was derived and subjected to actogram plotting. The plot pattern

was analyzed for assessment of circadian activity disruption.

- What is the circadian time used in the comparison shown in fig 2A ?

Response: The original time point was ZT6 when mRNA of Rev-erba reaches the peak. However, we agree with the reviewer that the data at a single time point is insufficient. To alleviate the reviewer's concern, for jet-lagged mice, we have performed new similar experiments at a different circadian time point of ZT18 (corresponding to a trough phase of Rev-erba mRNA). For Bmal1 knockout mice, we performed new experiments at six circadian time points. The new data have been added to new Figure 2A & B.

- In the result section, the 2.2 and 2.3 may be grouped together as there some redundancy between the two paragraphs.

Response: The reviewer raised a comment that result sections of 2.2 and 2.3 may be grouped together. We disagree with the reviewer on this point. In Result section 2.2, we investigated the effects of circadian perturbation (both jet lag and Bmal1 knockout) on colitis, and found that disruption of circadian clock exacerbated experimental colitis. In Result section of 2.3, we investigated the role of Rev-erba in colitis development, and showed that Rev-erba ablation sensitized mice to experimental colitis. Clearly, the two sections have different research focuses, showing distinct findings. Therefore, we believe that it is unnecessary to combine the two sections.

- In fig 5 E, I don't see a clear effect of the competitor oligo. EMSA is not quantitative, outdated technique and, of no use since a Chip experiment is shown. The same comment applies to fig 6H.

Response: We agree with the reviewer. Accordingly, the EMSA data have been moved to Supplemental materials.

- The sex of the animals used is not indicated, were they males, females or both ?

Response: We regret the sex of animals was not indicated in previous submission. All mice used in this study were male. This formation has been added to the Methods section and shown below.

“Eight to fourteen weeks-old mice (male) were used for in vivo experiments.”

- In p4 (top) of the introduction, authors should refer to Rev-erb a and b.

Response: It may be not clear to the reviewer that we have appropriately introduced Rev-erba and Rev-erbb in the second paragraph of INTRODUCTION section. It may be of no concern to talk about circadian rhythms first and then the core clock gene Rev-erb.

- Fig 6B-D does not include confocal microscopy as indicated in the text.

Response: Thanks for pointing this out. We have corrected it to “Figure 6B, C, E&F”.

Reviewers' comments:

Reviewer #1 (Remarks to the Author):

I do commend the authors for the detailed revisions and addressing every single point. I am happy with most of the revisions. couple of points need further clarification: I am not completely convinced that only DSS model is enough to answer the questions the authors tried to answer. While there is no need to conduct further experiments with a different model, the authors do need to acknowledge that as a potential shortcoming. Similarly, there is still dearth of how all the research data translates.

Reviewer #2 (Remarks to the Author):

The manuscript has been largely improved, and most criticisms have been addressed adequately. However, the following issues remain to be clarified:

General

1. Jet lag protocol. You mention you used the one that was initially designed and implemented by Filipowski et al (JNCI 2005). When was it started precisely in your studies? On the first day of DSS exposure? Before DSS? If so how many days/weeks? Was it continued during SR9009 treatment?
2. Did DSS and/or SR9009 modify circadian rhythms in activity and drinking behaviour? The induction of IL1beta by DSS and its limitation by SR9009 support some modifications of sleepiness/wakefulness/circadian coordination by DSS, while SR9009 could exert opposite effects. Please discuss whether and how such circadian alteration/re-inforcement could impact on your data.
3. Dose response studies were performed for the in vitro experiments, but not the in vivo ones. What was the rationale for the selection of the SR9009 dose for the in vivo experiments? Timing at ZT08 was chosen as being in coincidence with highest circadian expression of REV-ERB alpha protein, as supported in Figure S11. However, did DSS alter such pattern?
4. SR9009 was effective if treatment was started before and throughout DSS induction. No efficacy was seen if it was started simultaneously with DSS suggesting limited efficacy once the disease was active. Could this depend upon dose of SR9009? Could the effective timing be modified in the mice receiving DSS in the absence of prior circadian "re-inforcement" with SR9009?
5. It still was not clear to me, why there was a need for NK-kappaB RE on Nirp3 in order for effects of REV-ERB alpha or SR9009 to be seen?
6. Circadian time series should be analysed also with cosinor and a period of 24 or 12 h, both in normal and in colitic mice (cf Figure 1). The colitic mice have phase shifted rhythms, that need to be statistically documented. As a result, treatment at ZT 8 in "healthy" mice, and in those exposed to DSS do not correspond to the same molecular clock timing. This issue should be discussed in the Discussion, as a limitation of your work, while not detracting from the main findings.

Specific

Abstract, line 42: the protective effect is "lost in Nirp3-/-". Please add: "and in Rev-erb alpha -/-"

Introduction

P6 Line 88: remove "3)"

Results

P8, line 128.: there is now a need to rename Figure 3 as Figure 2 and vice versa to remain consistent with the text.

P 9, line 158: "rhythmicity in gene expression was dampened in Rev-erb-alpha KO...". But IL1beta and IL18 appear to be rhythmic with maximum shifted by ~12 h as compared to controls (Figure 3). Please discuss this finding, as well. Could it impact on SR9009 efficacy in Rev-erb alpha-/- mice, with this agent being possibly effective at a time differing from ZT08 ?

P11, lines 190-and subsequent ones: this is where SR9009 dose and timing could matter....

P13: It would be more accurate to speak of prevention than to relied of colitis with SR9009 same comment P17, lines 306-...)

Discussion

P14; lines 241-244: the lack of any in vivo dose and timing responses to SR9009 precludes from such strong conclusion, and represent a limitation of the study that should be acknowledged, and stimulate further work on the chronoprevention and chronotherapy of experimental colitis.

P16, line 275 :word after Nr1d-???

Methods

P19: The histopathological scoring procedure and method should be precisely described, and the meaning of each score specified,

Legends to Figures

Fig 1: (A): Please tell the reader how the comparison can be made (in line? In rows? Both?

Reviewer #3 (Remarks to the Author):

The authors have addressed most of my concerns and the manuscript has been improved.

I still have a concern about 2 points:

1. A minimal information about the western blot analyses should be provided in the method (or supp methods) section (dilution used for the antibodies, reagents, how the quantification was done ?, which software ? etc). In addition, Although it is stated in figure legends that WB were done 3 times in independent experiments and statistically tested, nowhere is shown the results of the quantifications. Why not include this in the supplemental material since it is available ?
2. The ROR experiment shown in supp Fig 9 is not convincing at all for 2 reasons: there is no positive control using a synthetic RORE and the fold induction is at max 1.2 which may be statistically significant but is most likely biologically not relevant. It's a pity that only RORa was tested because various ROR isoforms often behave differently depending of the response element context. Therefore, I do not agree with the conclusion that ROR does activate Nlrp3 as stated line 214. Only the SR1001 experiment should be kept but again, controls are missing.

NCOMMS-18-01185B

We wish to thank the reviewers for careful and valuable reviews. Below are our point-by-point responses.

Responses to reviewer #1

I do commend the authors for the detailed revisions and addressing every single point. I am happy with most of the revisions. couple of points need further clarification: I am not completely convinced that only DSS model is enough to answer the questions the authors tried to answer. While there is no need to conduct further experiments with a different model, the authors do need to acknowledge that as a potential shortcoming. Similarly, there is still dearth of how all the research data translates

Response: We agree with the reviewer that additional experiments with a different model may be helpful but unnecessary in current work.

It appeared that the reviewer is concerned about how the research data translate into potential clinical applications. We believe that it should be of little concern because the central finding of Rev-erb α as an integrator of biological clock and colitis development would most likely lead to novel therapeutics (targeting Rev-erb α) for management of colitis.

Responses to reviewer #2

The manuscript has been largely improved, and most criticisms have been addressed adequately. However, the following issues remain to be clarified:

General

1. Jet lag protocol. You mention you used the one that was initially designed and implemented by Filipinski et al (JNCI 2005). When was it started precisely in your studies? On the first day of DSS exposure? Before DSS? If so how many days/weeks? Was it continued during SR9009 treatment?

Response: We regret that our statements were not clear. These specific details about jet lag protocol have been provided by adding the following sentence in “**Results2.2**”.

“Mice were subjected to jet lag for 8 weeks before DSS feeding. “

The reviewer may be not clear that the jet-lagged mice were not subjected to SR9009 treatment in this particular experiment. Therefore, SR9009 treatment is not applicable.

2. Did DSS and/or SR9009 modify circadian rhythms in activity and drinking behaviour? The induction of IL1beta by DSS and its limitation by SR9009 support some modifications of sleepiness/wakefulness/circadian coordination by DSS, while SR9009 could exert opposite effects. Please discuss whether and how such circadian alteration/re-inforcement could impact on your data.

Response: The reviewer asked for a discussion about whether and how DSS/SR9009 modify circadian behaviors. Since current study aimed to investigate circadian regulation of colonic inflammation, it was reasonable to center our work on the inflammatory dissection other than other biological processes or animal behaviors.

To alleviate the reviewer’s concern, we have added following sentences (in DISCUSSION) to acknowledge the lack of explorations on circadian behaviors imposed by DSS/SR9009.

“In addition, DSS and SR9009 may modify the circadian behaviors^{43,44}. Whether and how such circadian alterations affect inflammation processes remain unexplored.”

3. Dose response studies were performed for the in vitro experiments, but not the in vivo ones. What was the rationale for the selection of the SR9009 dose for the in vivo experiments? Timing at ZT08 was chosen as being in coincidence with highest circadian expression of REV-ERB alpha protein, as supported in Figure S11. However, did DSS alter such pattern?

Response: The reviewer raised a question how the authors chose SR9009 dose for animal study. In fact, the SR9009 dose selection was appropriately made according to the original dose-response studies (ranging from 50-100 mg/kg) by Solt et al (Nature. 2012;485(7396):62-8). In particular, a relatively lower dose of 50 mg/kg was initially chosen in our studies to ensure least side-effects. Since a significant pharmacological effect (anti-colitis) was observed, this dose was maintained throughout the studies.

The reviewer raised a second question whether DSS alters circadian pattern of Rev-erb α and how this affects dosing-time selection (for SR9009). Based on our results, DSS (for colitis induction) did cause changes in Rev-erb α expression (Figure 1D). To be specific, Rev-erb α expression was down-regulated and its rhythmicity was significantly blunted (Figure 1D). Due to diminished Rev-erb α rhythmicity, we believe that the effect of DSS on dosing-time (for SR9009) was minimal, and that ZT8 should be a reasonable dosing-time (for SR9009) as the first step (proof-of-concept studies) to clarify the role of Rev-erb α in colitis development, and to elucidate the underlying mechanisms.

4. SR9009 was effective if treatment was started before and throughout DSS induction. No efficacy was seen if it was started simultaneously with DSS suggesting limited efficacy once the disease was active. Could this depend upon dose of SR9009? Could the effective timing be modified in the mice receiving DSS in the absence of prior circadian “re-inforcement” with SR9009?

Response: It appears that the reviewer is questioning why pre-treatment of SR9009 was performed in efficacy studies. The purpose of pre-treatment to ensure the steady-state concentrations (highest drug exposures, based on pharmacokinetic theory) and to maximize activation effects of Rev-erba prior to colitis induction. This may be necessary because SR9009 is concerned with poor pharmacokinetic properties (i.e., short elimination half-life and high clearance) (J Med Chem. 2013;56(11):4729-37) with low systemic exposure (or bioavailability).

The reviewer is correct that SR9009 is less effective when it was administered after DSS challenge (Supplemental Figure 10C). Although the exact reason was unknown, a lower drug systemic exposure may be a contributing factor. In this regard, a dose escalation may enhance the effect because of improved drug exposure as commented by the reviewer (drug exposure is positively correlated with drug dose based on the pharmacokinetic theory).

As also noted by the reviewer, the dosing timing of SR9009 was chosen as being in coincidence with the highest circadian expression of Rev-erba (comment 3). It is noteworthy that SR9009 dosing does not alter the circadian pattern in Rev-erba expression (PLoS One. 2016;11(3):e0151014). Therefore, we believe that the effective timing should be not modified in the mice receiving DSS.

5. It still was not clear to me, why there was a need for Nk-kappaB RE on Nirp3 in order for effects of REV-ERB alpha or SR9009 to be seen?

Response: We regret that the main finding (i.e., the regulation mechanism of Nlrp3 by Rev-erb α) is not clear to the reviewer. In fact, we provided strong evidence that REV-ERB α activation both directly and indirectly (via p65) repressed Nlrp3 transcription and decreased activation and release of IL-1 β and IL-18 to alleviate the colitis. First, sequence analysis suggested one RevRE (Rev-erb α response element) and two κ B sites (NF- κ B binding sites) within the 210-bp region (Figure 5D&E). Second, mutation of either RevRE (-1139/-1129 bp) or κ B sites attenuated but failed to abolish the transcriptional activity of Rev-erb α (Figure 5D). Only a mutation of both RevRE and κ B sites completely abrogated the transcriptional activity effects (Figure 5D). Third, a combination of QPCR, Western blots, Immunofluorescence analysis, luciferase reporter assay, EMSA and Chip strongly indicated a repressive role of Rev-erb α in p65 expression and NF- κ B signaling (Figure A-I).

It is noteworthy that direct and/or indirect mechanisms are necessary in transcriptional factor mediated gene regulation. For example, REV-ERB α represses IL-6 expression not only (directly) through a REV-ERB α binding motif but also (indirectly) through an NF- κ B binding motif (ScientificWorldJournal. 2014;2014:685854). The xenobiotic-response receptor CAR inhibits CYP7A1 and PEPCK through direct competition with HNF-4 for binding to the DR1 motif and indirect competition with HNF-4 for binding to common coactivators (J Biol Chem. 2006;281(21):14537-46). Therefore, we believe that it is NOT surprising that REV-ERB α acts on NLRP3 through both Rev-erb α response element (a direct action) and NF- κ B binding sites (an indirect action).

6. Circadian time series should be analysed also with cosinor and a period of 24 or 12 h, both in normal and in colitic mice (cf Figure 1). The colitic mice have phase shifted rhythms, that need to be statistically documented. As a result, treatment at ZT 8 in “healthy” mice, and in those exposed to DSS do not correspond to the same molecular clock timing. This issue should be discussed in the Discussion, as a limitation of your work, while not detracting from the main findings.

Response: We have followed the reviewer’s suggestion and derived the circadian parameters using the cosinor method (new data summarized in Supplementary Table 5).

We have also followed the reviewer’s suggestion and added a paragraph (the second last paragraph in DISCUSSION, shown below) to discuss the rationale for dosing time selection as well as the potential issue of inconsistent clock timing between normal and colitis mice.

“It was noteworthy that SR9009 was dosed at ZT8 in animal efficacy studies. This dosing time was chosen as being in coincidence with the highest expression of Rev-erba protein in normal mice (Supplemental Figure 10A). However, circadian timing system may be altered in mice with DSS-induced colitis as suggested by dysregulated clock genes (Figure 1). In particular, Rev-erba expression was down-regulated and its rhythmicity was significantly blunted (Figure 1D). It was therefore acknowledged that ZT8 perhaps was not the optimal dosing time for maximized efficacy in the disease model. Nevertheless, significant pharmacological effects elicited by SR9009 were sufficient to clarify the role of Rev-erba in colitis development. “

Specific

Abstract, line 42: the protective effect is “lost in Nirp3-/- “. Please add: “and in Rev-erb alpha -/-“

Response: Fixed. Thanks.

Introduction

P6 Line 88: remove “3”

Response: It may be not clear to the reviewer that “NOD-like receptor family pyrin domain containing 3” is the full name of NLRP3. Thus “3” should be kept.

Results

P8, line 128.: there is now a need to rename Figure 3 as Figure 2 and vice versa to remain consistent with the text.

Response: We have carefully checked and there may be no problem here.

P 9, line 158: “rhythmicity in gene expression was dampened in Rev-erb-alpha KO...”. But IL1beta and IL18 appear to be rhythmic with maximum shifted by ~12 h as compared to controls (Figure 3). Please discuss this finding, as well. Could it impact on SR9009 efficacy in Rev-erb alpha-/- mice, with this agent being possibly effective at a time differing from ZT08 ?

Response: We regret that our statements were not clear to the reviewer. We did not mean to say L1beta and IL18 were dampened genes. In revised version, we have changed “rhythmicity in gene expression was dampened in Rev-erb-alpha KO...” to “rhythmicity in *Nlrp3* expression was dampened in Rev-erb-alpha KO...”

We have followed the reviewer’s suggestion and have described and discussed the finding that IL1beta and IL18 rhythmicity were altered in Rev-erb-alpha KO mice in the revised manuscript (in DISCUSSION).

The reviewer asked for again a discussion about the dosing time selection for SR9009. As responded above, we have added a paragraph (the second last paragraph in DISCUSSION) to discuss the rationale for dosing time selection and the potential influencing factors. We believe that circadian Rev-erb α expression should be the key determinant to dosing time selection because Rev-erb α is the pharmacological target of SR9009.

P11, lines 190-and subsequent ones: this is where SR9009 dose and timing could matter....

Response: The reviewer may make a wrong judgment here. The luciferase reporter assays were in vitro experiments. The so-called “dosing time problem” is not applicable. The dose was appropriately selected based on previous dose-effect studies (Figure 4B).

P13: It would be more accurate to speak of prevention than to relied of colitis with SR9009 same comment P17, lines 306-...)

Response: Revised as suggested. Thanks.

Discussion

P14; lines 241-244: the lack of any in vivo dose and timing responses to SR9009 precludes from such strong conclusion, and represent a limitation of the study that should be acknowledged, and stimulate further work on the chronoprevention and chronotherapy of experimental colitis.

Response: We agree with the reviewer that further studies are needed to establish dose-efficacy relationships and dosing time-efficacy relationships (dosing regimen) for optimized therapeutics. Accordingly, we have followed the reviewer's suggested and added a statement to acknowledge this limitation (shown below).

"Further works are needed to establish optimal dose and dosing time for SR9009 in terms of drug development and clinical therapeutics."

P16, line 275 :word after Nr1d-???

Response: Fixed.

Methods

P19: The histopathological scoring procedure and method should be precisely described, and the meaning of each score specified,

Response: Revised as suggested. The details have been added (shown below).

“In brief, histological damage was scored based on goblet cells loss, mucosa thickening, inflammatory cells infiltration, submucosa cell infiltration, ulcers and crypt abscesses. A score of 1–3 or 1–4 were given for each parameter (scoring criteria provided in Supplemental Table 6) with a maximal total score of 20.”

Legends to Figures

Fig 1: (A): Please tell the reader how the comparison can be made (in line? In rows? Both?

Response: Revised as suggested. The following sentences have been added to the Legend of Figure 1A.

“Each column represents a gene and each row presents colon samples from different time points. Red indicates high relative expression and green indicates low expression of genes as shown in the scale bar.”

Responses to reviewer #3

The authors have addressed most of my concerns and the manuscript has been improved.

I still have a concern about 2 points:

1. A minimal information about the western blot analyses should be provided in the method (or supp methods) section (dilution used for the antibodies, reagents, how the quantification was done ?, which software ? etc). In addition, Although it is stated in figure legends that WB were done 3 times in independent experiments and

statistically tested, nowhere is shown the results of the quantifications. Why not include this in the supplemental material since it is available ?

Response: We have followed the reviewer's suggestion. Details of Western blotting method have been provided in "**Methods**" section.

We also have followed the reviewer's suggestion to include the quantification data in "**Supplemental material**" (please see new Supplemental Figure 12).

2. The ROR experiment shown in supp Fig 9 is not convincing at all for 2 reasons: there is no positive control using a synthetic RORE and the fold induction is at max 1.2 which may be statistically significant but is most likely biologically not relevant. It's a pity that only ROR α was tested because various ROR isoforms often behave differently depending on the response element context. Therefore, I do not agree with the conclusion that ROR does activate Nlrp3 as stated on line 214. Only the SR1001 experiment should be kept but again, controls are missing.

Response: The reviewer raised a comment that a positive control with a synthetic RORE should be used. The reviewer may be not very clear about this particular experiment. In this experiment, the Raw264.7 cells were transfected with ROR α (overexpression) plasmid. At the end of transfection, Nlrp3 and p65 mRNA expressions were determined by QPCR. To our best knowledge, a control of synthetic RORE may be not applicable to such experiments.

We accept the reviewer's criticism that it's premature to conclude that ROR α activates Nlrp3/p65 due to the lack of in vivo evidence (although our in vitro experiments support regulation of Nlrp3/p65 by ROR α). To alleviate the reviewer's concern, we

have removed the ROR experiment data and the relevant result descriptions. We believe that the ROR data may be of little help to the main theme of current study which focused on circadian regulation of colonic inflammation by Rev-erba and elucidations on the underlying mechanisms.

REVIEWERS' COMMENTS:

Reviewer #1 (Remarks to the Author):

Thanks for addressing all the comments in details

Reviewer #2 (Remarks to the Author):

My criticisms have been answered to in a satisfactory fashion